# Tumor matrix stiffness promotes metastatic cancer cell interaction with the endothelium

Steven E Reid[1], Emily J Kay[1], Lisa J Neilson[1], Anne-Theres Henze[2], Jens Serneels[2], Ewan J McGhee[1], Sandeep Dhayade[1], Colin Nixon[1], John BG Mackey[1,3], Alice Santi[1], Karthic Swaminathan[1], Dimitris Athineos[1], Vasileios Papalazarou[1,4], Francesca Patella[1], Álvaro Román-Fernández[5], Yasmin ElMaghloob[1], Juan Ramon Hernandez-Fernaud[1], Ralf H Adams[6], Shehab Ismail[1], David M Bryant[1,5], Manuel Salmeron-Sanchez[4], Laura M Machesky[1], Leo M Carlin[1] (iD), Karen Blyth[1], Massimiliano Mazzone[2,7] (iD) & Sara Zanivan[1,5,*] (iD)

## Abstract

Tumor progression alters the composition and physical properties of the extracellular matrix. Particularly, increased matrix stiffness has profound effects on tumor growth and metastasis. While endothelial cells are key players in cancer progression, the influence of tumor stiffness on the endothelium and the impact on metastasis is unknown. Through quantitative mass spectrometry, we find that the matricellular protein CCN1/CYR61 is highly regulated by stiffness in endothelial cells. We show that stiffness-induced CCN1 activates β-catenin nuclear translocation and signaling and that this contributes to upregulate N-cadherin levels on the surface of the endothelium, *in vitro*. This facilitates N-cadherin-dependent cancer cell–endothelium interaction. Using intravital imaging, we show that knockout of *Ccn1* in endothelial cells inhibits melanoma cancer cell binding to the blood vessels, a critical step in cancer cell transit through the vasculature to metastasize. Targeting stiffness-induced changes in the vasculature, such as CCN1, is therefore a potential yet unappreciated mechanism to impair metastasis.

**Keywords** CCN1/CYR61; stiffness; blood vessels; proteomics; cancer metastasis
**Subject Categories** Cancer; Cell Adhesion, Polarity & Cytoskeleton
**The EMBO Journal (2017) 36: 2373–2389**

## Introduction

Tumor cells and cancer-associated fibroblasts (CAFs) influence the extracellular matrix via increased matrix deposition and modification (Kalluri & Zeisberg, 2006; Pickup *et al*, 2014). The tumor matrix typically consists of excessive levels of fibrous collagen, which can be additionally crosslinked by soluble mediators such as lysyl oxidase (LOX), thereby elevating matrix stiffness (Butcher *et al*, 2009). In turn, the increased matrix stiffness can have a profound effect on cancer progression inducing oncogenic intracellular signaling to aid tumorigenesis, including activation of FAK, AKT, β-catenin, and PI3K, and inhibition of the tumor suppressor genes PTEN and GSK3α/β (Mouw *et al*, 2014). Increased tumor stiffness not only impacts cancer cells but its effects similarly extend to the surrounding stromal cells, where matrix stiffness can activate fibroblasts to a CAF phenotype and maintain it via the mechanosensitive transcription factor YAP (Georges *et al*, 2007; Calvo *et al*, 2013). In addition, matrix stiffness correlates with the number of tumor-activated macrophages (Acerbi *et al*, 2015). Therefore, tumor matrix stiffness is becoming an appealing target for therapeutic intervention (Jarvelainen *et al*, 2009; Cox & Erler, 2011). Targeting tumor stiffness via the inhibition of LOX activity has been shown to decrease tumor growth, malignancy, and metastasis in mice (Levental *et al*, 2009; Miller *et al*, 2015). Furthermore, solid stress within tumors was reduced with an angiotensin inhibitor, resulting in CAF inactivation and reduced collagen and hyaluronan levels, even in established tumor matrices (Chauhan *et al*, 2013). These studies support tumor microenvironment stiffness as a therapeutic target to perturb cancer development and progression.

1 Cancer Research UK Beatson Institute, Glasgow, UK
2 Lab of Tumor Inflammation and Angiogenesis, Center for Cancer Biology, VIB, Leuven, Belgium
3 Inflammation, Repair and Development, Imperial College London, London, UK
4 Division of Biomedical Engineering, School of Engineering, University of Glasgow, Glasgow, UK
5 Institute of Cancer Sciences, University of Glasgow, Glasgow, UK
6 Department of Tissue Morphogenesis, Faculty of Medicine, Max-Planck-Institute for Molecular Biomedicine, University of Münster, Münster, Germany
7 Lab of Tumor Inflammation and Angiogenesis, Center for Cancer Biology, Department of Oncology, KU Leuven, Leuven, Belgium
*Corresponding author. Tel: +44 141 330 3971; E-mail: s.zanivan@beatson.gla.ac.uk

A crucial element to cancer metastasis is the vascular endothelium. While normally acting as a physiological barrier, during tumorigenesis the vasculature provides a major route for cancer cell dissemination to distant sites. Despite the known consequences that tumor matrix stiffness has on cancer cells and fibroblasts to aid cancer metastasis, whether the matrix stiffness sets up a similar feedback with the endothelium to facilitate cancer cell entry and exit from the vasculature is a fundamental, yet unaddressed issue.

We sought to determine how primary human umbilical vein endothelial cells (HUVECs referred to as ECs) responded to matrices of low (400 Pa) or high (22,000 Pa) stiffness, using fibronectin-coated polyacrylamide gels (PAGs) that mimic normal or transformed tissue, respectively (Butcher *et al*, 2009; Leventhal *et al*, 2009).

## Results

### High stiffness induces phenotypic and cell signaling alterations in endothelial cells

ECs cultured at increasing levels of matrix stiffness displayed corresponding increases in proliferation, as assessed by EdU incorporation (Fig 1A), spread area (Fig 1B), and activation of MLC, FAK, AKT, and ERK1/2 (Fig 1C and D). These results recapitulated responses to stiffness in other studies (Ghosh *et al*, 2008; Wood *et al*, 2011) and validated our model of the endothelium response to differing matrix stiffness.

To assess how the proteome of ECs is affected by tumor stiffness, we combined the most accurate quantitative approach, Stable Isotope Labeling of Amino acids in cell Culture (SILAC) (Ong *et al*, 2002), with high-resolution mass spectrometry (MS). SILAC-labeled ECs were cultured on PAGs of differing matrix stiffness (physiological, 400 Pa, and pathological, 22,000 Pa) for 24 h before MS analysis (Fig 1E). MS analysis quantified 5,938 proteins, with 5,461 of these with a calculated ratio between low stiffness and high stiffness in replicate experiments, where the heavy and light labeling conditions were swapped (i.e., forward and reverse experiments; Dataset EV1). We further pinpointed 244 proteins most highly regulated in both the forward and reverse experiments (Fig EV1A and Dataset EV1). Next, we used STRING to map the functional and physical protein–protein interactions (Fig EV1B) and to perform an unbiased pathway enrichment analysis of the regulated proteins. This analysis observed upregulation of proteins related to cell adhesion, metabolism, and proliferation (Fig EV1C), in response to increased stiffness, which mirrored the observed phenotype. Of note, cell adhesion proteins included receptors involved in heterotypic cell–cell interactions (Fig EV1D), indicating that matrix stiffness may regulate the crosstalk between different cell types within the tumor microenvironment.

As proliferation is a major response to increased stiffness (Fig 1A and D), we segregated changes induced by matrix stiffness from those induced by proliferation by comparing the proteomes of ECs in different proliferative states seeded onto substrates of the same stiffness (Fig 1F). This allowed us to subtract proliferation-associated proteins to focus on the stiffness-related proteome of ECs (Fig 1G and Dataset EV2). One of the most stiffness-regulated proteins was the secreted protein CCN1, which is known as a pro-angiogenic factor that binds to integrins, thus regulating intracellular signaling from the extracellular matrix (Leu *et al*, 2002). CCN1 possessed characteristics of a matricellular protein, being found in cell lysates, secreted medium, and extracellular matrix extracts from ECs, and its levels increased in all fractions when cells were stimulated with high matrix stiffness (Appendix Fig S1A). Finally, inhibition of myosin with blebbistatin treatment reduced CCN1 protein and mRNA levels at low stiffness and high stiffness, and blocked the increase in CCN1 expression in response to high matrix stiffness (Fig 1H and I). Similarly, we measured reduced levels of stiffness-induced CCN1 in microvascular dermal endothelial cells (HMVECs) upon blebbistatin treatment (Appendix Fig S1B). This indicates that CCN1 levels are regulated by cytoskeletal tension and that elevated levels of CCN1 are, at least in part, a *bona fide* response to matrix stiffness in ECs.

Next, we assessed that a link between CCN1 and tumor stiffness exists *in vivo* using the orthotopically transplanted mouse E0771 breast cancer cell model. *In situ* hybridization analysis revealed that Ccn1 was expressed in cancer and stromal cells, including blood vessels (Fig 2A). High Ccn1 expression was found only in some regions of the tumor which were adjacent to the necrotic areas (Appendix Fig S1C). Quantification of collagen I and III fibers in tumor regions with high (peri-necrotic areas) or low Ccn1 expression by Sirius red staining showed that higher collagen content associated with high Ccn1 expressing regions (Fig 2A and B). Finally, atomic force microscopy analysis of the tissues determined that peri-necrotic tumor areas (*bona fide* highly expressing Ccn1) were much stiffer than the non-necrotic areas and that stiffness was within a range comparable to those recapitulated *in vitro* with the PAGs (Fig 2C). Hence, an association between CCN1 and stiffness can be found *in vivo* and we investigated further the role of CCN1 in endothelial cells in the tumor context.

### CCN1 regulates N-cadherin expression

First, we deciphered the mechanism by which CCN1 may affect endothelial cells and that could be relevant in a tumor context. Notably, efficient silencing of CCN1 in ECs using a pool of siRNA minimally altered the stiffness-induced proliferation and spreading of ECs after 24 h of culture (Fig EV2A–C). This suggests that at least in our system, CCN1 does not function as a general regulator of proliferation or cell–matrix adhesion. We reasoned that CCN1 may be part of the signaling response of cells to increased stiffness. Therefore, using MS and a SILAC spike-in approach (Geiger *et al*, 2011), we investigated whether any of the stiffness-regulated proteins were also regulated downstream of CCN1 (Fig 3A and Dataset EV3). Strikingly, the levels of few proteins followed CCN1 regulation. Among these, N-cadherin/CDH2, which is notable as a cancer cell protein that promotes metastasis (Tanaka *et al*, 2010), was highly downregulated in response to CCN1 silencing (Fig 3A). Corroborating the above results, both CCN1 and N-cadherin levels were induced by stiffness in HMVECs (Fig EV2D and E), and CCN1 silencing with either a pool of or a single siRNA markedly reduced the upregulation of N-cadherin induced by high matrix stiffness in ECs and HMVECs (Figs 3B and EV2D–G). The stiffness-induced CCN1-dependent induction of N-cadherin was specific, as the levels of the major endothelial cadherin, VE-cadherin/CDH5, were

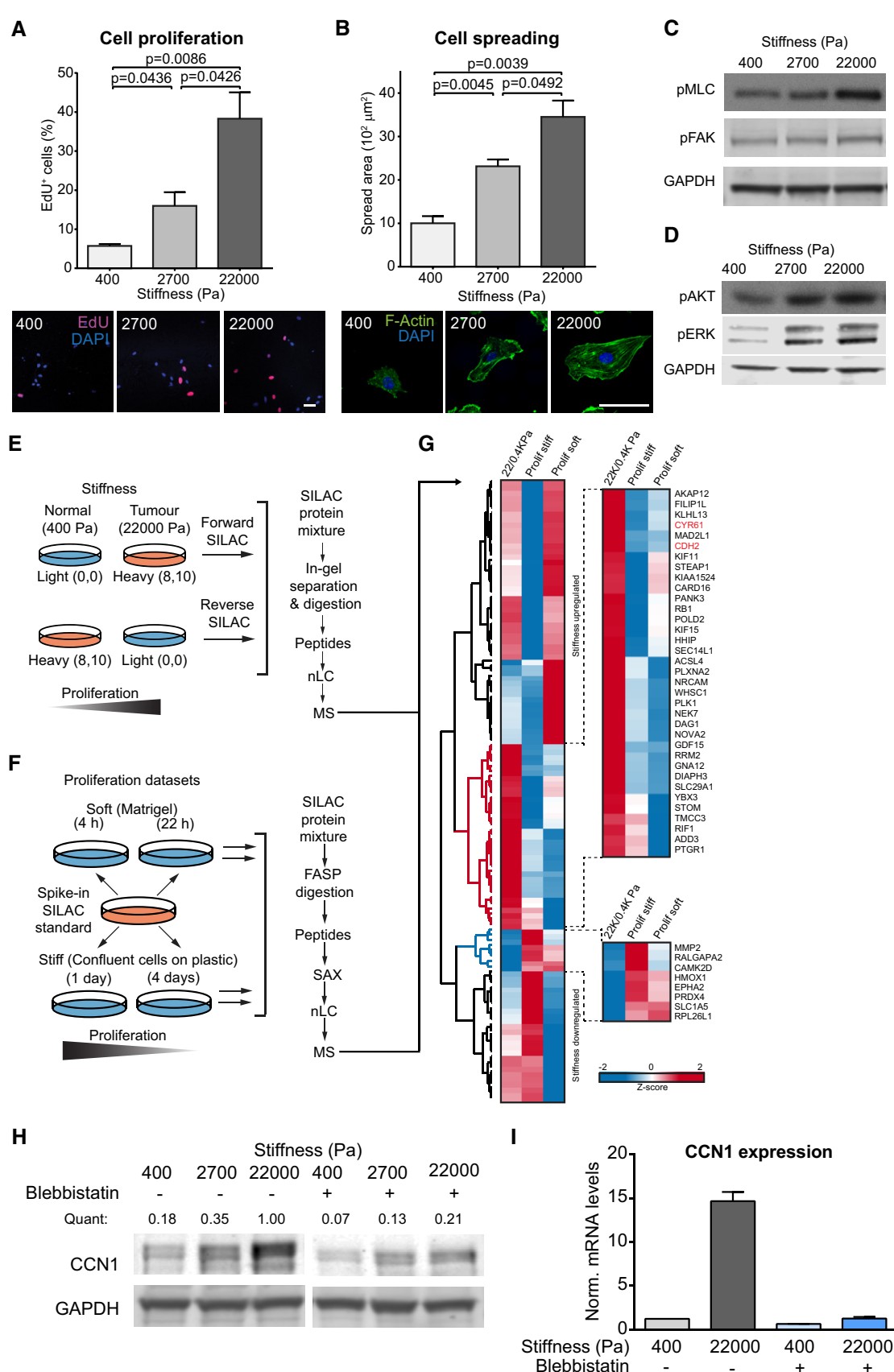

**Figure 1.**

**Figure 1.** **CCN1 levels are highly induced by matrix stiffness in endothelial cells.**

A, B   HUVECs cultured for 24 h on FN-coated PAGs of differing matrix stiffness have increased proliferation, measured as percentage of cells which incorporated EdU (3 h of incubation, Click-IT EdU Imaging kit, Invitrogen) (A), and have increased spread area (B). Bars represent mean ± SEM. N = 3 average values from 3 replicate experiments. Scale bar = 50 μm. Significance based on two-tailed unpaired *t*-test.

C, D   Representative Western blot analysis showing that the intracellular signaling is altered in HUVECs with increasing matrix stiffness, as shown by active phosphorylated myosin light chain (pMLC, Thr18, Ser19), active phospho-FAK (pFAK, Tyr397), active phospho-AKT (pAKT, Ser473) as well as activated phospho-ERK1 and phospho-ERK2 (pERK1, Thr202, Tyr204; pERK2, Thr185, Tyr187). GAPDH was used as a loading control.

E   Schematic of the SILAC labeling approach where the heavy and light conditions are mixed to generate forward and reverse replicate experiments before being processed for MS analysis.

F   In the proliferation datasets, a heavy labeled SILAC standard was spiked into each sample and used as a reference. FASP = filter-aided sample preparation. SAX = strong anion exchange.

G   Hierarchical clustering based on average Euclidean distance and heat map based on the Z-score of the $\log_2$ averaged ratios of proteins regulated in HUVECs by stiffness and that were quantified in the proliferation datasets (Dataset EV2). The red and blue clusters contain proteins whose levels were found more up- or downregulated, respectively, by stiffness compared with the proliferation datasets. 22/0.4 KPa = average SILAC ratio between forward and reverse experiments of the stiffness datasets; Prolif soft = average SILAC ratio of three replicate experiments of HUVECs cultured on Matrigel: 4 h/22 h; Prolif stiff = average SILAC ratio of three replicates of HUVECs cultured on plastic: 1 day/4 days.

H   HUVECs pre-treated with blebbistatin were seeded on FN-coated PAGs of different stiffness for 24 h in the presence of blebbistatin. Representative Western blot showing CCN1 levels and GAPDH was used as a loading control. Quant = CCN1 intensity normalized by GAPDH intensity (analysis by Image Studio Lite software).

I   HUVECs pre-treated with blebbistatin were seeded on FN-coated PAGs of different stiffness for 24 h in the presence of blebbistatin. Representative RT–PCR analysis showing Ccn1 expression normalized by three housekeeping genes. Bars represent mean ± SEM. n = 3 technical replicates.

Source data are available online for this figure.

unaffected by stiffness or CCN1 loss (Fig 3B). Stiffness and CCN1 have been shown to induce epithelial to mesenchymal transition (EMT; Hou *et al*, 2014; Wei *et al*, 2015). However, here, ECs are not undergoing EMT since the levels of the endothelial cell markers, VE-cadherin, PECAM-1, and von Willebrand factor, were not altered, as measured in the MS analyses. Furthermore, only N-cadherin was increased, but not other typical mesenchymal markers, such as Vimentin (Dataset EV3). The induction of N-cadherin expression occurred as a transcriptional response to CCN1, as CCN1 silencing in ECs significantly reduced N-cadherin mRNA levels to a degree that mirrored protein levels (Fig 3C). A corresponding increase in N-cadherin protein and mRNA levels occurred upon CCN1-GFP overexpression (Fig 3D and E), where ectopic GFP-tagged CCN1 was abundantly expressed, secreted into the media, and similarly localized to endogenous CCN1 in the matrix (Fig EV2H and I). The EC response to increased stiffness thus involves CCN1-dependent upregulation of N-cadherin expression.

### CCN1-induced expression of N-cadherin requires β-catenin

We sought to determine how CCN1 controls N-cadherin upregulation. We examined β-catenin (CTNNB1) as it has been shown to be stiffness sensitive (Samuel *et al*, 2011), involved in CCN1-dependent signaling (Xie *et al*, 2004) and regulates N-cadherin expression (Lamouille *et al*, 2014). First, we ascertained that stiffness-induced CCN1 controlled β-catenin levels. The levels of β-catenin increased with increasing matrix stiffness in ECs while silencing CCN1 strongly reduced the levels of β-catenin, as well as N-cadherin (Figs 4A and B, and EV2F). Next, we confirmed that β-catenin transcriptional activity increases with matrix stiffness (Fig 4C), and showed that stiffness-induced N-cadherin required β-catenin. In ECs silenced for β-catenin, we measured similar levels of N-cadherin when cultured at low stiffness and high stiffness (Appendix Fig S2A). To focus specifically on CCN1-dependent signaling, we manipulated CCN1 levels in ECs grown at a constant stiffness (on plastic). CCN1 knockdown reduced the transcriptional activity of β-catenin (Fig 4D). Conversely, overexpression of CCN1-GFP induced increased β-catenin nuclear

localization (Fig 4E and Appendix Fig S2B and C) and further elevation in β-catenin transcriptional activity (Fig 4F). This was associated with an increase in both N-cadherin and β-catenin levels, and in particular active β-catenin (non-phosphorylated on S37/T41; Maher *et al*, 2010; Fig 4G). This indicates that CCN1 is a major intermediate in stiffness-induced β-catenin nuclear translocation and transcriptional activity. Next, we explored the requirement of β-catenin for the CCN1-dependent regulation of N-cadherin expression. In CCN1-GFP-overexpressing cells, knockdown of β-catenin to control levels diminished the ability of the cells to induce upregulation of CCN1-induced N-cadherin, both protein and mRNA, to the levels of the control cells (Fig 4G and H, and Appendix Fig S2D). Endothelial CCN1 therefore modulates N-cadherin expression in a β-catenin-dependent manner.

### Cancer cell adhesion to the endothelium increases with matrix stiffness and requires CCN1-induced N-cadherin

N-cadherin has a major role in mediating cellular contact during the initial attachment of cancer cells to the endothelium. This heterotypic interaction is the first step of the metastatic cascade which leads to the subsequent transendothelial migration of the cancer cells (Qi *et al*, 2005; Strell *et al*, 2007). For this reason, we addressed whether CCN1-dependent N-cadherin upregulation in ECs could facilitate cancer cell interactions with ECs. First, we evaluated N-cadherin localization in ECs cultured at differing stiffnesses. N-cadherin localization to peripheral cell–cell contacts was consistently increased with matrix stiffness (Fig 5A and Appendix Fig S3A and B). Moreover, the increase in N-cadherin surface localization induced by elevated stiffness was strongly reduced upon CCN1 knockdown (Fig 5A and Appendix Fig S3A and B). In contrast, the localization of VE-cadherin at the cell periphery was not affected by stiffness or by CCN1 knockdown (Fig 5A and Appendix Fig S3A and B). Concordantly, silencing of CCN1 in a monolayer of ECs did not alter its permeability (Appendix Fig S3C). To simulate the initial process of cancer cell intravasation, we measured the adhesion of cancer cells to a confluent monolayer of ECs, where the lowest stiffness was replaced with 1,050 Pa. At this

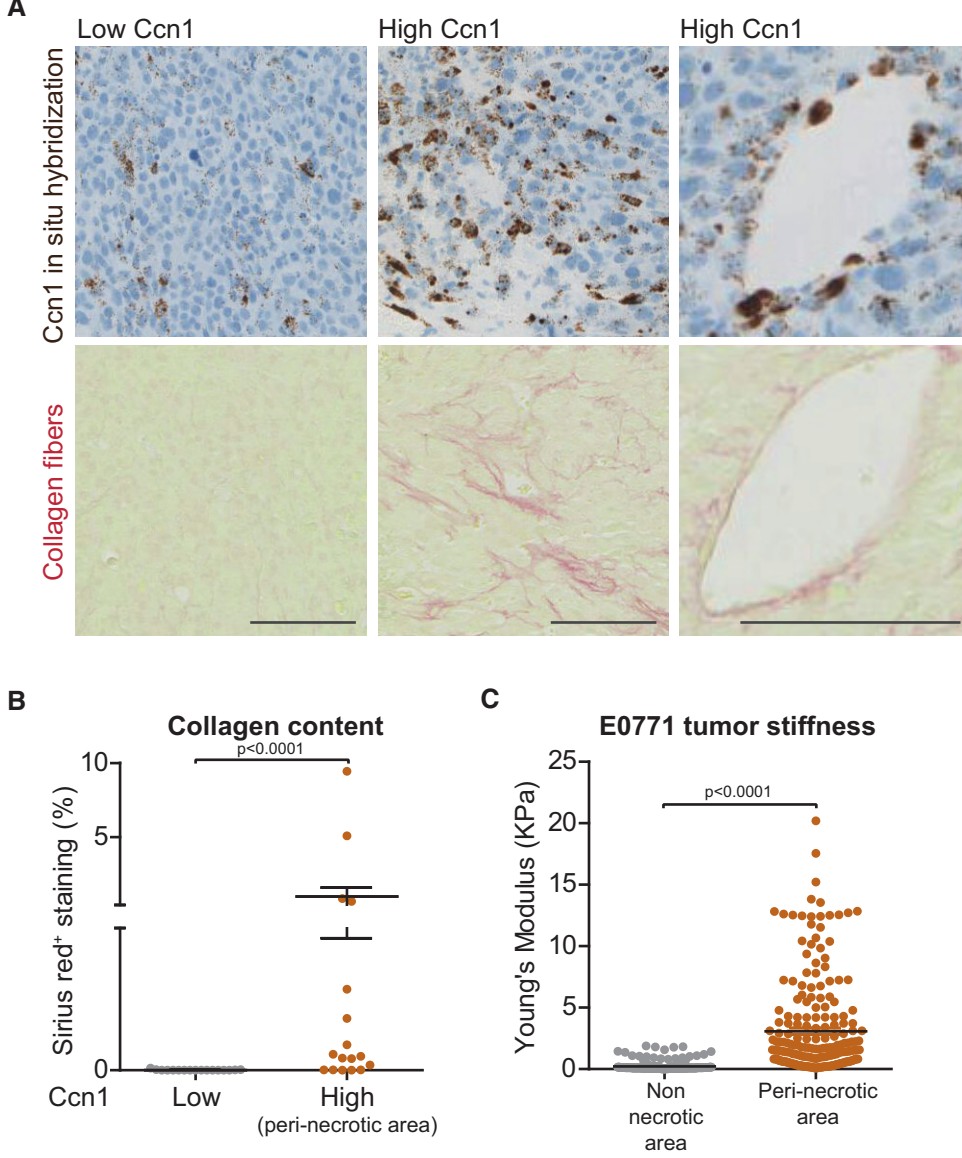

**Figure 2.  Ccn1 is highly expressed in stiff regions of orthotopic E0771 tumors.**

A   Representative image of *in situ* hybridization for Ccn1 mRNA and collagen I and III fibers (Sirius red) of E0771 orthotopic tumors performed on consecutive sections showing that there are tumor regions with high and low expression of Ccn1. The right panels show a blood vessel positive for Ccn1 staining. Scale bar = 100 μm.

B   Quantification of collagen I and III fiber content based on Sirius red staining (% of the measured area) showing that higher amounts of collagen fibers are found in regions of the tumor expressing high Ccn1 levels.

C   Quantification of tumor stiffness performed by atomic force microscopy showing that higher stiffness is measured in peri-necrotic regions (*bona fide* expressing high Ccn1 levels) of the tumor.

Data information: Significance according to two-tailed Mann–Whitney test. Horizontal lines and error bars represent mean ± SEM. *n* = regions assessed from two tumors.

stiffness, cells are able to form greater substrate adhesions (Saunders & Hammer, 2010) and they formed an intact monolayer, while at 400 Pa we could not observe the formation of an intact monolayer. We used the PC3 human prostate cancer line, which has high N-cadherin levels and has been utilized previously in co-culture with ECs (Nalla *et al*, 2011). PC3 cell adhesion to ECs increased with matrix stiffness, which could be significantly abrogated by CCN1 knockdown (Fig 5B and Appendix Fig S3D). Even at the highest

levels of stiffness (culture on glass), CCN1 knockdown with a pool of siRNA or two independent siRNAs in ECs significantly reduced the adhesion of PC3 cells (Fig 5C and Appendix Fig S3E). The general validity of these results was confirmed using B16F1 mouse melanoma, Lewis lung carcinoma (LLC), and E0771 breast cancer cell lines, which express N-cadherin when kept in culture (Fig 5D and Appendix Fig S3F). Moreover, PC3 cells adhered less to HMVECs and ECs expressing lower levels of N-cadherin, where

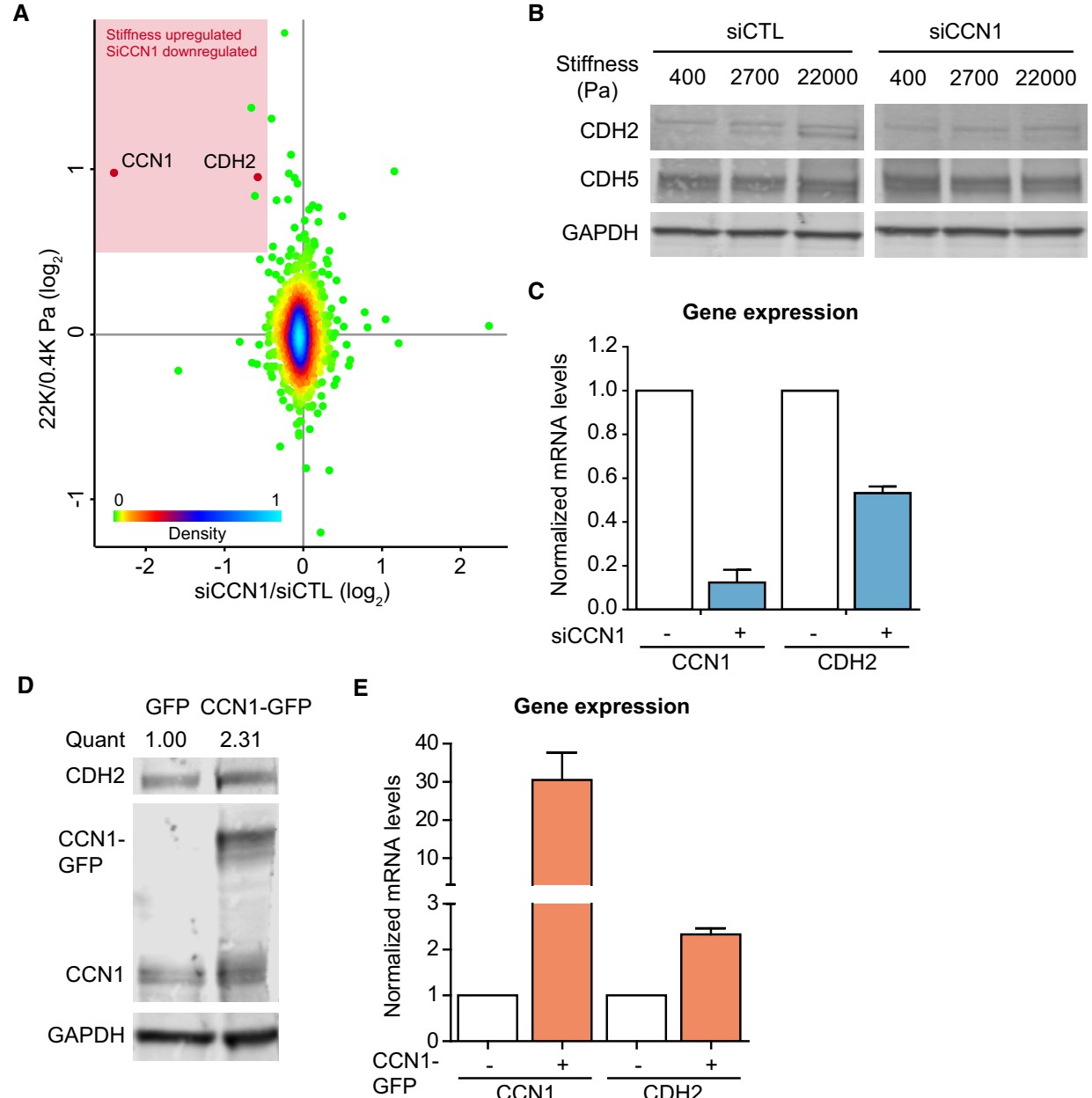

**Figure 3. CCN1 induces N-cadherin expression in HUVECs.**

A  Scatter plot showing the comparison between proteomic changes induced in HUVECs upon knockdown of CCN1 (siCCN1) and high stiffness. The x-axis reports the average ratio (n = 3 replicate experiments) (log$_2$) between siCCN1 and siCTL; the y-axis reports the average ratio (n = 2 experiments, forward and reverse) (log$_2$) between high stiffness and low stiffness. In the red panel are highlighted proteins that were highly upregulated by stiffness and downregulated in CCN1-silenced cells. The colors of the dots represent the density distribution (0 = 0% and 100 = 100%) of the SILAC ratios calculated between the two experiments (Perseus analysis).

B  Representative Western blot analysis of CCN1 knockdown in HUVECs cultured on FN-coated PAGs of different stiffness shows that N-cadherin (CDH2), but not VE-cadherin (CDH5) is upregulated by stiffness and reduced upon CCN1 silencing.

C  RT–PCR showing that N-cadherin mRNA levels are reduced upon CCN1 knockdown in HUVECs. Bars represent mean ± SEM. n = 3 experimental replicates.

D  Representative Western blot analysis shows that N-cadherin protein levels increase with CCN1 overexpression in HUVECs.

E  RT–PCR showing CCN1 overexpression in HUVECs increases N-cadherin mRNA levels. Bars represent mean ± SEM. n = 3 experimental replicates.

reduced levels of N-cadherin were induced upon silencing of CCN1 (Fig 5D and Appendix Fig S3G) or β-catenin (Fig 5E and Appendix Fig S3H), respectively. Conversely, exogenous expression of CCN1-GFP in ECs strongly enhanced PC3 cell adhesion (Fig 5F

and Appendix Fig S3I) and addition of an N-cadherin-blocking antibody completely abrogated the ability of CCN1 to promote EC–cancer cell interactions (Fig 5F). Moreover, stable knockdown of N-cadherin in PC3 cells strongly decreased their adhesion to control

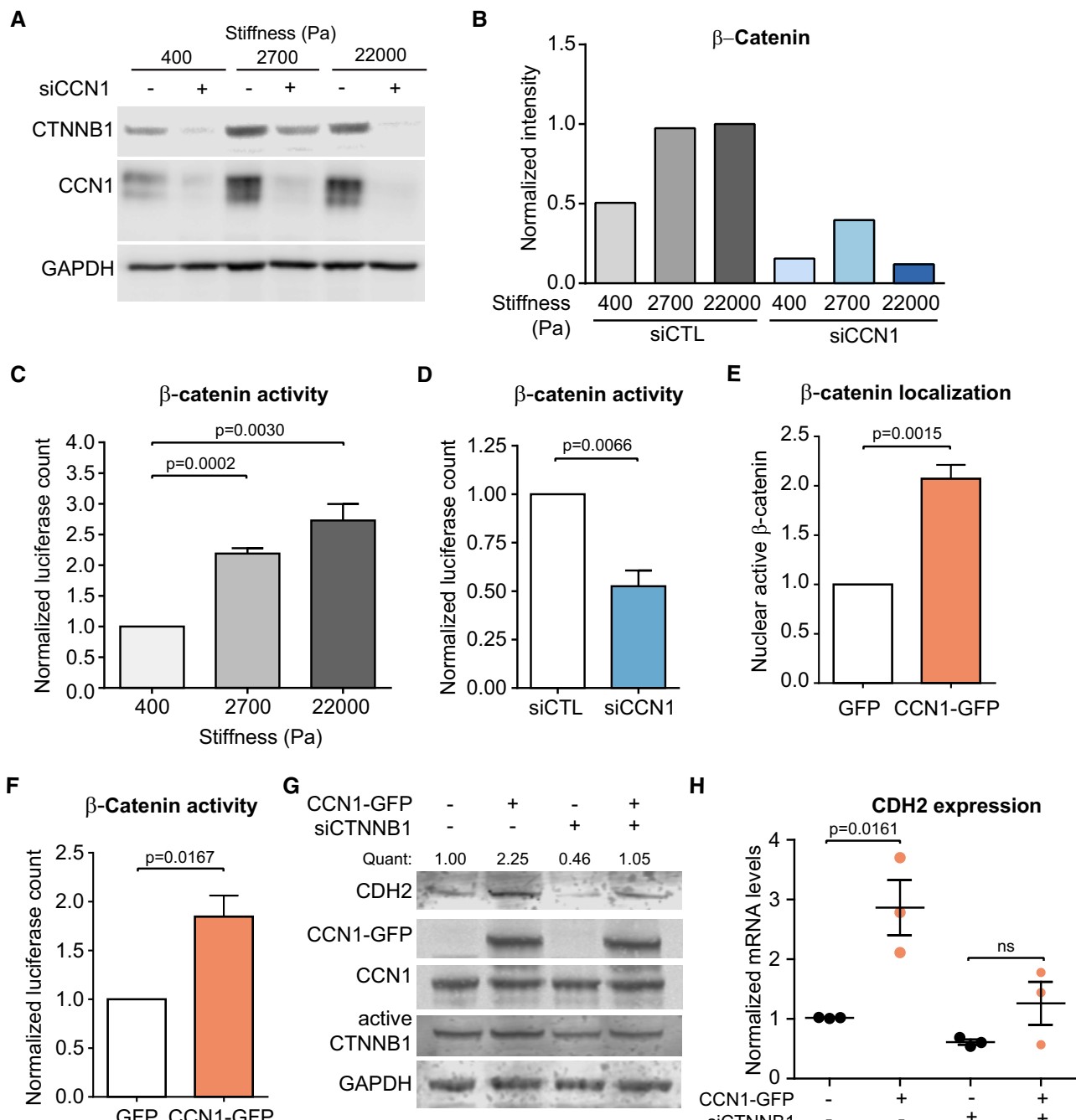

**Figure 4. CCN1-induced N-cadherin regulation requires β-catenin.**

A, B    Representative Western blot analysis (A) and quantification (B) showing that upon silencing of CCN1 in HUVECs (single siRNA, same Western blot shown in
Fig EV2F) cultured on FN-coated PAGs of different stiffness, the levels of β-catenin (CTNNB1) are induced by stiffness to a lower extent. β-catenin levels were
normalized by GAPDH, which was used as a loading control. Quantification based on Image Studio Lite software.
C, D    HUVECs cultured on plastic and transfected with a β-catenin luciferase reporter indicate that β-catenin activity increases with matrix stiffness (C) and decreases
upon CCN1 knockdown (D).
E, F    Overexpression of CCN1-GFP increases β-catenin nuclear localization (E) and transcriptional activity (F).
G       Representative Western blot analysis showing that the levels of N-cadherin increase with the overexpression of CCN1-GFP in ECs and that this was largely reduced
upon knockdown of β-catenin. Quant = CDH2 intensity over GAPDH intensity, which was used as a loading control. Quantification based on Image Studio Lite
analysis.
H       RT–PCR analysis showing that the levels of N-cadherin increase with the overexpression of CCN1-GFP in HUVECs and that this was largely reduced upon
knockdown of β-catenin. The levels of N-cadherin were normalized to the levels of three housekeeping genes.

Data information: Bars represent mean $\pm$ SEM ($n$ = 3 replicate experiments). Significance according to the unpaired Student's *t*-test.
Source data are available online for this figure.

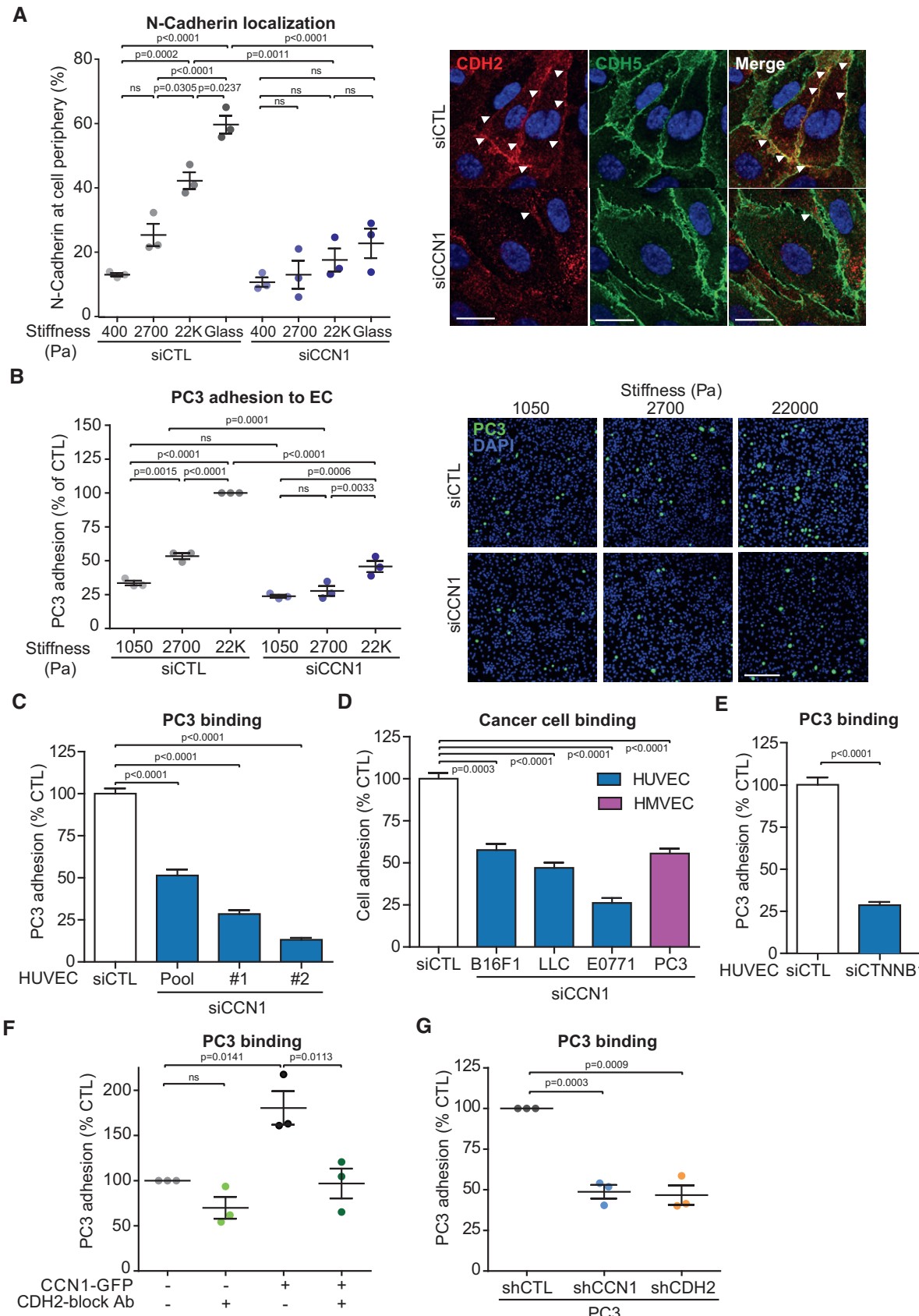

**Figure 5.**

**Figure 5. Endothelial N-cadherin promotes interaction with cancer cells.**

A   Representative immunofluorescence analysis and quantification for N-cadherin (CDH2) in HUVECs cultured for 24 h on FN-coated PAGs or FN-coated glass showing that the localization at the cell periphery increases with elevated stiffness, which was reduced with CCN1 knockdown. White arrowheads highlight localization of CDH2 at the cell periphery. VE-cadherin (CDH5) staining is not altered in CCN1-silenced cells. Scale bar = 20 μm.

B   Representative immunofluorescence and quantification analysis showing that more PC3 cells adhere to HUVECs cultured on FN-coated PAGs of high matrix stiffness; however, CCN1 knockdown in the HUVECs largely prevents this increased adhesion. Scale bar = 200 μm.

C   Silencing of CCN1 with a pool of siRNA (pool) or two single siRNAs (#1 and #2) in HUVECs reduces the binding of PC3 cells. siCTL $n = 168$, siPool $n = 18$, si#1 $n = 83$, si#2 $n = 89$ fields measured from three independent experiments, each performed in three technical replicates. The significance for each siCCN1 was calculated against each specific control.

D   Silencing of CCN1 with a pool of siRNA in HUVECs or HMVECs reduces the binding of B16F1 melanoma, Lewis lung carcinoma (LLC), E0771 breast cancer, and PC3 cells. CTL $n = 248$, LLC/E0771 $n = 90$, PC3 $n = 70$ fields measured from three independent experiments, each performed in three technical replicates. B16F1 $n = 3$ averaged measurements from three independent experiments. The significance for each cell line was calculated against their specific control.

E   Silencing of β-catenin (CTNNB1) with a pool of siRNA in HUVECs reduces the binding of PC3 cells. $N = 90$ fields assessed from three independent experiments, each performed in three technical replicates.

F   Overexpression of CCN1-GFP in HUVECs increases PC3 adhesion, which is ablated in the presence of an antibody that functionally blocks the homophilic interaction of N-cadherin (CDH2).

G   Stable shRNA expression in PC3 cells targeting either CCN1 or CDH2 decreases their binding to control HUVECs.

Data information: Data are represented as mean ± SEM. For panels (A, B, and F), significance according to one-way ANOVA with Tukey's test for multiple comparisons ($n = 3$ replicate experiments). For panels (C–E and G), significance according to two-tailed unpaired *t*-test. ns: non-significant.

ECs (Fig 5G and Appendix Fig S3J). This indicates that sufficient levels of N-cadherin in both ECs and cancer cells must be present for EC–cancer cell interactions. Given this requirement for homophilic N-cadherin interactions in heterotypic cell interactions, we tested the generality of whether CCN1 also controlled N-cadherin expression in tumor cells and fibroblasts. Culture of immortalized mammary normal (iNF) or pro-invasive cancer-associated fibroblasts (iCAF; Kojima *et al*, 2010; Hernandez-Fernaud *et al*, 2017) at increasing stiffness resulted in a corresponding increase in CCN1 levels (Appendix Fig S3K). When we cultured these cells at high stiffness, CCN1 depletion resulted in downregulation of N-cadherin in iNFs, iCAFs, and PC3 cells (Appendix Fig S3L). Accordingly, stable (Fig 5G and Appendix Fig S3M) and transient (Appendix Fig S3N) CCN1 depletion in PC3 cells resulted in strong perturbation of adhesion to ECs (Fig 5G and Appendix Fig S3O), to levels similar to N-cadherin depletion (Fig 5G and Appendix Fig S3J). These data show that increased stiffness in the tumor microenvironment may cause increased CCN1 expression and cell–cell interactions between cancer cells, endothelial cells, and fibroblasts and that this depends on N-cadherin.

## Ccn1 knockout in the endothelium decreases cancer cell binding to the blood vessels

We examined the requirement of CCN1 in the endothelium for the highly metastatic B16F10 melanoma model (Fidler, 1975) to adhere to blood vessels *in vivo*. Since endothelial deletion of *Ccn1* postnatally can alter vessel growth in the developing mouse retina (Chintala *et al*, 2015), we crossed *Ccn1*$^{fl/fl}$ mice (Fig EV3A) with endothelial-specific *Cdh5Cre*$^{ERT2}$ driver mice (Wang *et al*, 2010) to delete *Ccn1* in adult mice (referred to as *Ccn1 KO*$^{EC}$). We ascertained that *Ccn1* was efficiently knocked out in the endothelium of the *Ccn1 KO*$^{EC}$ mice and that this did not affect the vasculature. Lungs of Ccn1 wild-type mice (*Ccn1 WT*$^{EC}$) positively stained for Ccn1 mRNA and the staining was restricted to some regions of the lungs (Fig EV3B and C). Ccn1 has been found expressed mostly in angiogenic endothelial cells (Chintala *et al*, 2015), suggesting that Ccn1 is expressed only in the regions of the lung that are undergoing active vascular remodeling. Conversely, no staining was detected in the lungs of *Ccn1 KO*$^{EC}$ mice (Fig EV3B and C). We could not detect significant differences in the lung vasculature between *Ccn1 KO*$^{EC}$ and *Ccn1 WT* mice, as measured by total amount of Pecam1$^+$ staining (Fig EV3D and E). Similarly, endothelial knock out of Ccn1 reduced the levels of Ccn1 expression in the ear (Fig EV3F). Moreover, Ccn1 deletion reduced the expression of N-cadherin in the lung vasculature (Fig EV3G), indicating that, also *in vivo*, CCN1 may control N-cadherin expression. To assess cancer cell binding to the vasculature, fluorescently labeled B16F10 melanoma cells were intradermally transplanted into the ear of *Ccn1 KO*$^{EC}$ and *Ccn1 WT*$^{EC}$ mice. We accurately monitored the capability of the cancer cells to adhere to the blood vessels by fluorescently labeling the vasculature with an anti-Pecam1 antibody. Intravital imaging analysis revealed that cancer cells can stably or transiently bind to blood vessels and, strikingly, the number of cancer cells that stably adhered to the blood vessels was significantly reduced upon depletion of Ccn1 in the endothelium (Fig 6A and B, and Movies EV1 and EV2). Hence, also *in vivo* endothelial Ccn1 regulates the crosstalk between cancer and endothelial cells by promoting cancer cell binding. Next, we provide evidence that vascular Ccn1 controls cancer cell metastasis. To localize the knockout of Ccn1 within the vascular regions, we exploited a cell-penetrating, soluble form of Cre, fused to the His-TAT-Nuclear localization sequence tag (HTNC). The HTNC recombines loxP sites when used in cell culture (Peitz *et al*, 2002), and targets endothelial cells when injected in the circulation of mice (Giacobini *et al*, 2014). *Ex vivo* addition of HTNC to mouse lung endothelial cells isolated from *Ccn1*$^{fl/fl}$ mice reduced Ccn1 protein levels (Appendix Fig S4A). Moreover, serial intravenous treatment with HTNC of B16F10-transplanted *Rosa26*$^{flSTOP-tdRFP}$ mice (Luche *et al*, 2007) induced efficient recombination in the vasculature of some tumor regions, as shown by immunohistochemistry staining for RFP (Appendix Fig S4B and C). Little recombination was measured in the lungs (Appendix Fig S4B and C) and no recombination in circulating myeloid cells, as assessed by the absence of RFP-positive signal in cells isolated from blood and bone marrow (Appendix Fig S4D). Furthermore, a strong staining for Ccn1 was measured in blood vessels of the B16F10 tumors (Appendix Fig S4E), which highlights the relevance of this tumor model for our study. *Ccn1*$^{fl/fl}$ mice were serially treated with HTNC (Fig 6C).

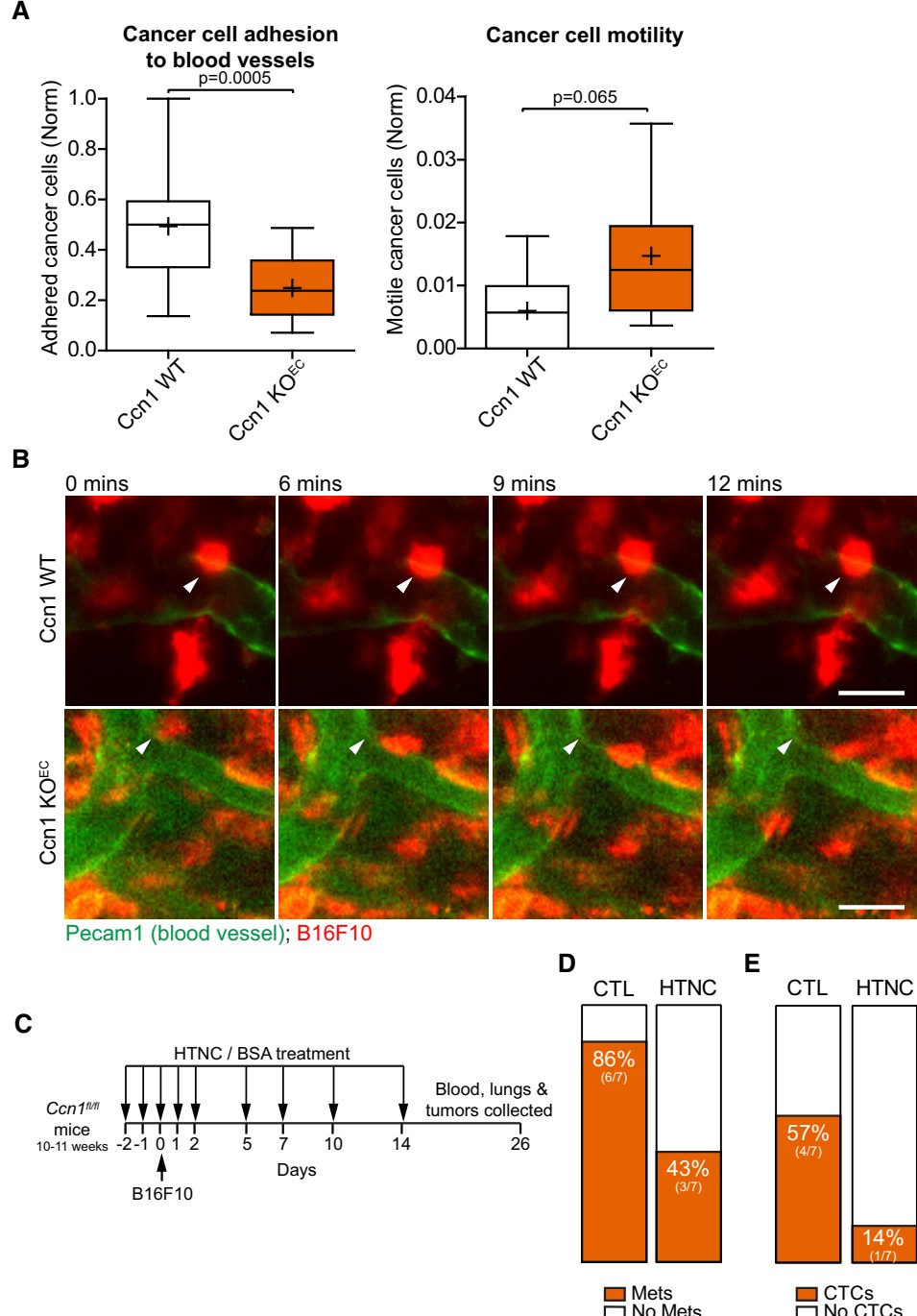

**Figure 6. Ccn1 endothelial knockout reduces cancer cell adhesion to blood vessels and attenuates metastasis.**

A Box plot (1st and 3rd quartile, the line represents the median; whiskers: min to max) of intravital imaging analysis of B16F10 melanoma cells injected intradermally in the mouse ear showing that less cancer cells adhere to and are more motile on blood vessels depleted for Ccn1 in the endothelial cells (*Ccn1 KO$^{EC}$*) compared with control mice expressing normal levels of Ccn1 in the vasculature (*Ccn1 WT*). Significance according to two-tailed unpaired *t*-test. WT left plot *n* = 16, KO left plot *n* = 17, WT right plot *n* = 16, KO right plot *n* = 17 fields measured from 7 mice/group. + = mean.

B Representative images of the intravital imaging analysis showing examples of fluorescently labeled B16F10 cells which stably adhere to the blood vessel of the ear when endothelial Ccn1 is expressed (*Ccn1 WT*), while transiently adhere when endothelial Ccn1 is knocked out (*Ccn1 KO$^{EC}$*). Blood vessels were stained by intravenous injection of fluorescently labeled anti-Pecam1. The arrowhead in each panel indicates the position of the highlighted cell at time 0 min. Scale bar = 20 μm.

C The treatment regime indicates when either HTNC or the BSA, which was used as a control, was administered by tail vein injection relative to the B16F10 subcutaneous implantation.

D Incidence of macrometastases in the lungs was decreased by HTNC treatment. *N* = 7 mice for each treatment group.

E The proportion of mice with detectable MLANA in the blood (= circulating tumor cells, CTCs) was decreased with HTNC treatment. *N* = 7 mice for each treatment group.

HTNC treatment had no effect on primary tumor growth, tumor vascularization, vasculature pericyte coverage, or tumor hypoxia (Appendix Fig S5A–F), indicating that partial depletion of *Ccn1* in the vascular regions did not alter the formation and function of the tumor-associated vasculature. In contrast, HTNC treatment suppressed the ability of the B16F10 cells to colonize the lung, as evidenced by a reduced incidence of macrometastases in lung tissue and detectable circulating tumor cells (Fig 6D and E). These data strongly suggest a defect in cancer cell transit into the blood stream, likely due to altered cancer cell entry into the vasculature. Supporting this hypothesis, we could detect almost no recombination in the lungs of HTNC-treated mice (see above). Moreover, we assessed that endothelial Ccn1 had no impact on metastasis formation in the lungs using the endothelial-specific *Ccn1* KO setup. Injection of B16F10 cells in the circulation of *Ccn1 KO^{EC}* and *Ccn1 WT^{EC}* mice did not show any significance decrease in lung colonization (Appendix Fig S5G and H) and formation of metastasis in the lungs (Appendix Fig S5I) in the absence of endothelial Ccn1. *In vitro* data corroborated the role of endothelial CCN1 in cancer cell intravasation. When PC3 cells were let to adhere for 24 h onto a monolayer of ECs, a lower degree of disruption of the endothelial monolayer was measured when ECs were silenced for CCN1 (Fig 7A and B, and Appendix Fig S6A). Finally, we showed that silencing of CCN1 in HMVECs did not impair the permeability of these cells when assembled into a monolayer, while it reduced the passage of highly invasive PC-3 TEM4-18 cells (Drake *et al*, 2009; Fig 7C and D, and Appendix Fig S6B and C). Thus, we propose that endothelial loss of CCN1 decreases cancer cell adhesion to blood vessels and reduces metastasis, likely by regulating cancer cell entry into the vasculature.

## Discussion

The formation of metastasis relies on the cancer cells' ability to escape from the primary tumor site, bind to blood and lymphatic vessels, intravasate, and exploit the blood flow to reach and colonize distant sites. High tumor stiffness closely correlates with disease progression and can drive invasion. Here we discovered that tumor stiffness alters the CCN1/β-catenin/N-cadherin pathway that contributes to the metastatic cascade by facilitating the binding of the cancer cells to the blood vessels. A model of our findings integrated in the cancer cell metastatic process is shown in Fig 7E.

We have explored the effect that the tumor matrix stiffness may have on the resident and attracted endothelial cells within the tumor environment. For this, we have used a range of physiological and pathological stiffnesses which have been shown to influence the endothelial phenotype. While at low physiological stiffness, such as 400 Pa, endothelial cells form capillary-like networks in 2D and tubules in 3D matrix, at increasing stiffness endothelial cells assemble into networks with larger lumens and fewer branches (Sieminski *et al*, 2004; Saunders & Hammer, 2010). Analysis of the proteome of ECs on hydrogels of physiological and pathological stiffness revealed that ECs respond to high stiffness by upregulating hundreds of proteins. Of those, a subset of 32, which includes CCN1, has been previously shown by chromatin immunoprecipitation assay to be under the control of mechanosensitive YAP/TAZ in

MDA-MB-231 breast cancer cells (Zanconato *et al*, 2015; Dataset EV1). Conversely, the majority of proteins have not yet been related to stiffness and provides a resource of proteins to be further investigated in this context.

We demonstrate that CCN1 induces enhanced nuclear β-catenin localization and transcriptional activity in endothelial cells and that a key downstream effect is N-cadherin expression. It has been previously reported that N-cadherin co-localizes with VE-cadherin at intercellular junctions and that it acts upstream of VE-cadherin in HUVECs and microvascular dermal endothelial cells. Indeed, silencing of N-cadherin results in increased endothelial permeability (Luo & Radice, 2005). While we could also observe a clear co-localization between VE-cadherin and N-cadherin in HUVECs, the residual levels of N-cadherin measured upon silencing of CCN1 did not affect the levels and localization of VE-cadherin and the permeability of the cells. We envisioned that in a tumor context, the CCN1-regulated N-cadherin levels control the heterotypic interaction between cancer cells and the endothelium to aid the promotion of intravasation and metastasis. N-cadherin is critical for EC attachment of melanoma and breast cancer cells and subsequent transendothelial migration (Qi *et al*, 2005; Strell *et al*, 2007). Ectopic N-cadherin expression in a non-metastatic prostate cancer model induces metastatic behavior in these cells, while blocking antibodies to N-cadherin reduced tumor growth, invasion, and metastasis, thus demonstrating the clinical relevance of N-cadherin interactions (Tanaka *et al*, 2010). Recently, high stiffness has been shown to induce N-cadherin expression in smooth muscle cells and fibroblasts (Mui *et al*, 2015), and we show that this extends to the endothelial cells. We build on this by characterizing a molecular mechanism within ECs, whereby the tumor-induced matrix stiffness increases CCN1 levels, which can modulate β-catenin signaling and consequently N-cadherin levels.

CCN1 is aberrantly expressed in many cancer types and high levels associate with tumor aggressiveness and metastasis (Xie *et al*, 2001; Sun *et al*, 2008). In a model of lung metastasis where MDA-MB-231 breast cancer cells are intravenously injected, CCN1 depletion in the cancer cells inhibited metastasis by reducing extravasation to the lung and enhancing cancer cell anoikis (Huang *et al*, 2017). Here we identified another mechanism through which CCN1 plays key roles in the metastatic cascade, by mediating the crosstalk between blood vessels and cancer cells. We show that levels of endothelial CCN1 determine the binding between endothelial and cancer cells *in vivo* and *in vitro* using co-culture assays. Our data indicate that CCN1 controls intravasation and not extravasation. Indeed, while we found decreased incidence of metastasis in mice partially depleted for Ccn1 in the tumor vasculature, there were no differences in lung metastasis between *Ccn1 WT* and *Ccn1 KO^{EC}* mice using a tail vein injection setup. However, the expression levels of both Ccn1 and N-cadherin were rather low in the lungs of non-tumor-bearing mice. To rule out that endothelial Ccn1 has no roles in cancer cell extravasation at the metastatic sites, further studies are required to investigate Ccn1 regulation in the vasculature of pre-metastatic organs.

We propose that clinical targeting of CCN1 may be beneficial in reducing tumor growth, invasion, and angiogenesis, and we demonstrate that targeting CCN1 in the vasculature, which is easily accessible to therapeutics, could also prevent cancer intravasation and subsequent metastasis.

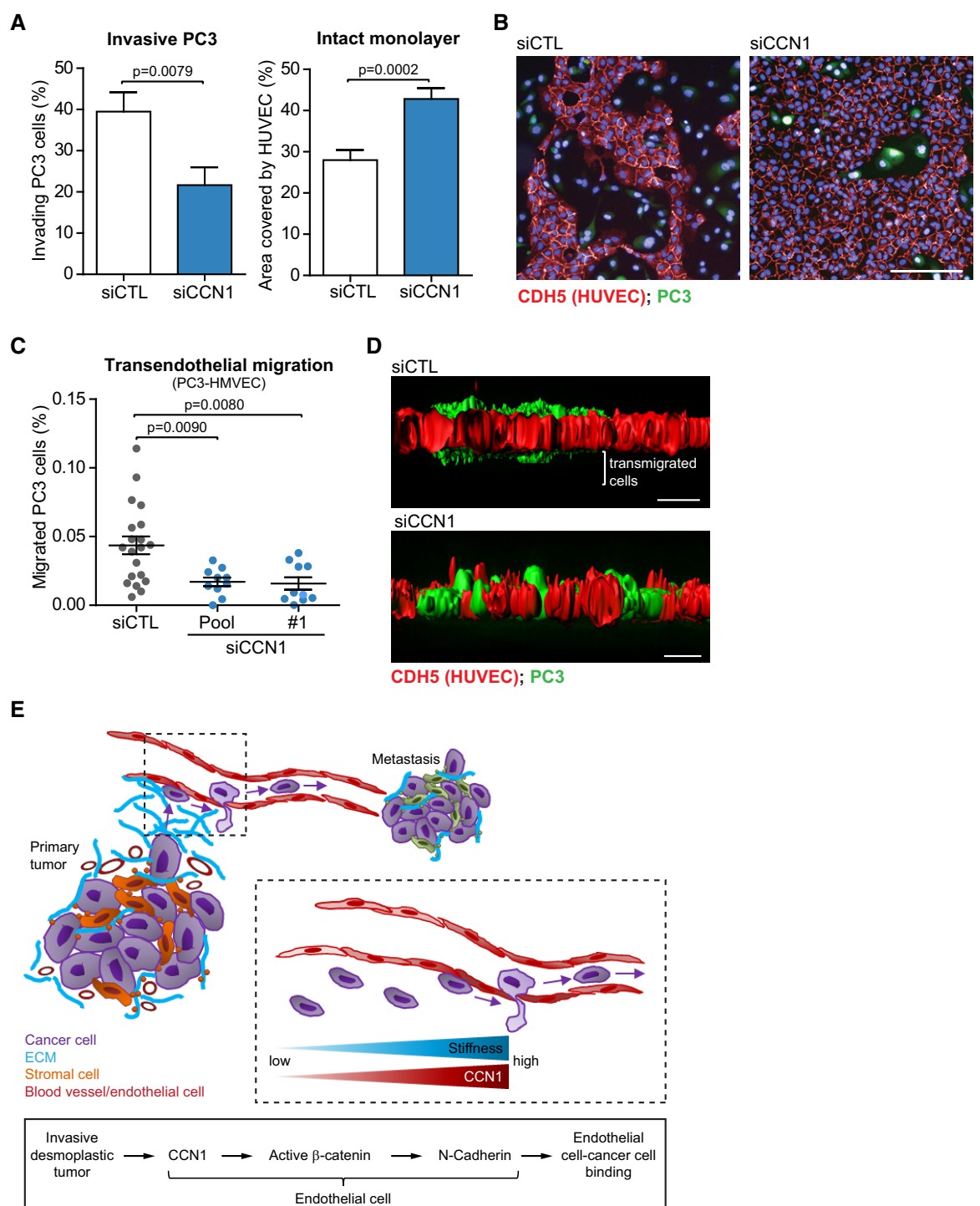

**Figure 7.  Reduced Ccn1 levels in endothelial cells inhibit cancer cell transendothelial migration.**

A   Silencing CCN1 in a monolayer of HUVECs inhibits the capability of PC3 cells to invade the endothelial monolayer, as measured by reduced number of PC3 cells that invaded the monolayer after 24 h of co-culture (left), and reduced amount of disrupted regions of the monolayer (right). siCTL $n$ = 21, siCCN1 $n$ = 23 fields assessed from one representative experiment of two.

B   Representative immunofluorescence staining of (A). Red = VE-cadherin (CDH5, HUVECs); green = fluorescently labeled PC3 cells. Scale bar = 100 μm.

C   Transendothelial migration assay performed with PC3 cells (clone TEM4-18) and HMVECs on transwells showing that silencing of CCN1 in HMVECs reduces the passage of cancer cells through the endothelial monolayer. $N$ = 20 (siCTL), 9 (siCCN1 pool) and 10 (siCCN1 #1) fields assessed from two independent experiments.

D   Representative immunofluorescence staining of (C). Red = VE-cadherin (CDH5, HUVECs); green = fluorescently labeled PC3 cells. Scale bar = 20 μm.

E   Working model for CCN1 in the tumor vasculature.

Data information: Bars represent mean ± SEM. Significance based on two-tailed unpaired $t$-test.

# Materials and Methods

### Cell culture

Human umbilical vein endothelial cells were isolated from freshly collected umbilical cords using collagenase (Roche Diagnostics) 0.2% in EBM-2 media (Lonza). Cells from 3 to 5 cords were pooled. Human adult dermal microvascular endothelial cells (HMVECs) were purchased from Sigma (100-05a). HUVECs and HMVECs were cultured in EGM-2 (Lonza) or EGM-2 10% FBS and no heparin, respectively, and used for experiments up to passage six. Immortalized human mammary normal fibroblasts (iNF) and cancer-associated fibroblast (iCAF) cell lines (kindly provided by Professor Akira Orimo; Kojima *et al*, 2010) and the melanoma cancer cell lines, B16F1, B16F10, and B16F10-GFP[+] (from the CRUK Beatson Institute), were cultured in DMEM (Gibco, Thermo Fisher Scientific), 10% FBS, 2 mM glutamine. The prostate cancer cell line PC3 was grown in RPMI medium (Gibco, Thermo Fisher Scientific) with 10% FBS, 2 mM glutamine. The E0771 breast cancer cells were purchased from Tebu-Bio and cultured in RPMI with 10% FBS. The LLC cells were purchased from Sigma-Aldrich and cultured in DMEM with 10% FBS. All cell lines were harvested with trypsin (0.025% in PBS + EDTA) and grown at 37°C, 5% $CO_2$, 21% $O_2$. Blebbistatin was from TOCRIS. All cell lines are routinely tested for mycoplasma. For primary antibodies, RNAi, and primers, see Appendix Supplementary Methods.

### Polyacrylamide gels for cell culture

Polyacrylamide gels were generated and coated with fibronectin as described in the literature (Lakins *et al*, 2012), with minor modifications: the functionalization solution differed from the published formulation by containing final concentrations: 0.0025% (v/v) di(trimethylolpropane) tetraacrylate and 0.01% (w/v) bisacrylamide, and acrylic acid *N*-hydroxysuccinimide ester (Sigma A8060) was used in place of *N*-succinimidyl acrylamidohexanoic acid.

### MS sample preparation

For the stiffness proteome, HUVECs were SILAC-labeled in custom-made EGM-2 medium without arginine and lysine (Lonza) and supplemented with $^{13}C_6{}^{15}N_2$ L-lysine and $^{13}C_6{}^{15}N_4$ L-arginine (heavy, Cambridge Isotope Laboratories) or L-lysine and L-arginine (light, Sigma) amino acids. Forward and reverse experiments were performed, where labeling conditions were swapped. For the proteome of HUVECs silenced for CCN1 and those cultured on Matrigel, a heavy SILAC-HUVEC standard was mixed at a 1:1 ratio with each of the lysates from the non-labeled samples (triplicates for each condition). Cell lysates were collected in 2% SDS, 100 mM Tris–HCl pH 7.6, subsequently reduced with DTT, and boiled before being run through a 4–12% gradient NuPAGE Novex Bis-Tris gel (Life technologies) in MOPS running buffer (Life technologies). Proteins were digested in-gel (stiffness and siCCN1 experiments) (Shevchenko *et al*, 2006), or digested on filter using the FASP protocol with trypsin (Promega) and peptides separated into six fractions using on-tip strong anion exchange (SAX) chromatography (Wisniewski *et al*, 2009) (Matrigel experiment). Digested peptides were desalted using StageTip (Rappsilber *et al*, 2007). After removal of acetonitrile (ACN) using speed vacuum, peptides were resuspended in 1% TFA and 0.2% acetic acid buffer for MS data analysis.

### MS analysis

Digested peptides were injected on an EASY-nLC system coupled on line to a LTQ-Orbitrap Elite via a nanoelectrospray ion source (Thermo Scientific). Peptides were separated using a 20-cm fused silica emitter (New Objective) packed in house with reversed-phase Reprosil Pur Basic 1.9 μm (Dr. Maisch GmbH). MS data were acquired using the Xcalibur software (Thermo Scientific) and .raw files processed with the MaxQuant computational platform (Cox & Mann, 2008) and searched with the Andromeda search engine (Cox *et al*, 2011) against the human UniProt Consortium (2010) database (release-2012 01, 88,847 entries). See also Appendix Supplementary Methods.

### MS comparison to proliferation datasets

Mass spectrometry datasets of ECs in different contexts were used for comparison with the stiffness dataset. One model uses ECs cultured on plastic at high seeding density, which are proliferative at day 1 but at day 4 become tightly confluent and reduce proliferation to a basal rate (Patella *et al*, 2016). Another model cultures ECs on soft Matrigel for 4 or 22 h, where cells are initially proliferative and begin to undergo morphogenesis, while at later time points ECs stop proliferating and are within a quiescent network (Patella *et al*, 2015). ECs are therefore either cultured on matrices of very high or very low stiffness as a monolayer or a network, respectively. Together, these models encompass proteins regulated with proliferation independent of the stiffness and the matrix, as well as differences in morphogenesis and cell–cell contacts. For these two datasets, we used a SILAC spike-in approach (Geiger *et al*, 2011).

In the stiffness dataset, proteins were considered regulated when their calculated SILAC ratio was above 1 standard deviation (STD) from the average of all the SILAC ratio calculated in both the forward and reverse replicate experiments (Fig EV1A). The subset of significantly regulated proteins was analyzed with STRING (version 10.0; Szklarczyk *et al*, 2015). The protein–protein interaction network was built using text mining, experiments, and databases as evidence, and a minimum interaction score of 0.7 was required. The functional enrichment analysis was performed with STRING using the default parameters and the whole genome as statistical background. Data were visualized with Cytoscape (Cline *et al*, 2007). The hierarchical clustering analysis (Fig 1G) was performed with Perseus software (Tyanova *et al*, 2016) using *Z*-scored SILAC ratios. The SILAC ratios represent the average of at least two replicate experiments for each experimental condition.

### Ethical approval of animal studies

All mouse procedures were in accordance with ethical approval from University of Glasgow under the revised Animal (Scientific Procedures) Act 1986 and the EU Directive 2010/63/EU authorized through Home Office Approval (Project license number 60/4181 and 70/8645) and, for the HTNC experiment with *Ccn1*[fl/fl] mice, approved by the Institutional Animal Care and Research Advisory Committee of the K.U. Leuven.

## E0771 orthotopic tumors for stiffness analysis

Twelve-week-old female C57BL/6 mice were orthotopically injected in the fat pad with $0.5 \times 10^6$ E0771 breast cancer cells in 50 μl solution 1:1, medium:Matrigel (Corning). Tumors were harvested at end point (max size). Half of each tumor was frozen and used for atomic force microscopy analysis and hematoxylin and eosin staining, while the other half was formalin-fixed and paraffin-embedded and processed for hematoxylin and eosin staining, Sirius red, 0.1% Direct Red (41496LH, Sigma), 0.1% Fast Green (FCF, S142-2, Raymond Lamb), diluted 1:9 in picric acid (84512.260, vWR), and *in situ* hybridization staining according to the manufacturer instruction. For atomic force microscopy measurement, see Appendix Supplementary Methods.

## Endothelial-specific Ccn1 conditional KO mice

C57BL/6 *Cdh5-Cre^ERT2* mice (Sorensen *et al*, 2009; Wang *et al*, 2010) were bred into C57BL/6J (N6) mice carrying a *loxP*-flanked *Ccn1* gene (*Ccn1^fl/fl*, Fig EV3A) and further backcrossed to N8 generation. To induce Cre activity in adult, mice were given one intraperitoneal injection of tamoxifen (2 mg, in sunflower seed oil) daily for 3 days. For the generation of endothelial Ccn1 expressing (*Ccn1 WT*) and depleted (*Ccn1 KO^EC*) mice, *Cdh5-Cre^ERT2;Ccn1^+/+* and *Cdh5-Cre^ERT2;Ccn1^fl/fl* mice were used, respectively. The phenotype of endothelial Ccn1 depletion was assessed by *in situ* hybridization for Ccn1 in the lungs. Tail vein metastasis experiments with B16F10 cells are described in the Appendix Supplementary Methods.

## Intravital imaging

Twelve-week-old female and male C57BL/6J *Ccn1 WT* or *Ccn1 KO^EC* were intradermally transplanted in the ear with $1 \times 10^6$ B16F10 melanoma cells, of which 67% were unlabeled and 33% were either GFP$^+$ or labeled with DiD cell-labeling solution (V22887, Life Technologies) in 10 μl PBS. Two days post-transplantation mice were injected intravenously with a 50 μl of fluorescently labeled Pecam1 antibody (0.5 mg/ml). Then, anesthetic (10 mg/ml Hypnorm, 5 mg/ml Hypnovel, and water, ratio 1:1:6) was administered intraperitoneally at 10 μl/g mouse. After 10–15 min from injection, hairs were removed from the ears using depilatory cream (Veet®) and mice positioned on a heated stage insert containing a coverslip kept at 37°C. The ear was gently flattened against the coverslip and held in place with tape. Intravital imaging analysis was performed using a multiphoton microscope system (LaVision Biotec Trimscope 2) with a Coherent Chameleon Ultra 2 ti:sapphire laser which can be used for excitation wavelengths between 700 and 1,040 nm. A 25× water immersion objective with a numerical aperture of 1.0 (Zeiss) was used, which had been modified to incorporate a water chamber to facilitate long-term imaging in an inverted geometry. Two cooled PMT detectors were used with a 595LP dichroic (Chroma) spectrally filtering the emission into green and red channels which were further filtered using band-pass emission filters, 525/50 nm and 650/100 nm (Semrock), respectively. Z-stack images were taken of about 100 μm in depth with slices at 2-μm intervals, every 90 s for 20 min. After imaging analysis, the animals were culled by cervical dislocation and lungs and ears dissected for analysis. For intravital imaging analysis, see Appendix Supplementary Methods.

## HTNC experiments

Seven-week-old male C57BL/6 *Rosa26^flSTOP-tdRFP* mice were injected with either 200 μg HTNC/Tat-Cre peptide (for protein purification, see Appendix Supplementary Methods) or BSA in 20 mM HEPES and 0.6 mM NaCl by tail vein injection at days −3, −1, 2, 4, 6, and 10 relative to subcutaneous injection of $1.5 \times 10^5$ B16F10 melanoma cells. Mice were culled 2 weeks post-injection. To assess the recombination in lungs and tumors, tissues were formalin-fixed and paraffin-embedded and sections stained for RFP. Two mice per group were used. One of the two mice injected with HTNC showed recombination in the tumor vasculature and was used for the analysis. For flow cytometry analysis, blood was collected by cardiac puncture. Bone marrow cells were isolated by flushing one femur and tibia from each mouse with approximately 5 and 2.5 ml, respectively, of RPMI/10% fetal bovine serum (FBS)/2 mM ethylenediaminetetraacetic acid (EDTA). Cells were passed through a 70-μm cell strainer forming single cell suspensions prior to red blood cell (RBC) lysis (8.3 g NH$_4$Cl, 1.0 g KHCO$_3$, 37.2 mg Na$_2$EDTA, 1 l dH$_2$O, pH 7.2–7.4) and flow cytometry analysis (see Appendix Supplementary Methods).

Ten- to eleven-week-old female and male C57BL/6 homozygous *Cyr61/Ccn1^fl/fl* mice were injected with either 200 μg HTNC/Tat-Cre protein or BSA in 20 mM HEPES and 0.6 mM NaCl by tail vein injection at days −2, −1, 0, 1, 2, 5, 7, 10, and 14 relative to subcutaneous injection of $1.5 \times 10^5$ B16F10 melanoma cells. Mice were treated with pimonidazole on day 26 before being culled, and the primary tumor, lungs, and blood were collected for analysis (*n* = 7 mice for each treatment group). Tumor volume was determined by the formula: volume = $\pi*[d^2*D]/6$, where d and D are the minor and major tumor axes, respectively. Circulating tumor cells were isolated from the blood, and the RNA was extracted for the analysis of melanin A by RT–PCR. Whole tumor sections were imaged as a tile scan and were analyzed by a macro used in ImageJ. The channels were split before the individual images were put together into individual mosaic images for each channel and subsequently merged. Regions of interest were included or excluded from the analysis generating a "zone". Total Meca32 (green) and co-localization with Ng2 (red) within the zone were measured.

## Statistical analysis

Statistical analysis was carried out using GraphPad Prism software (GraphPad Software, Inc.). The *P*-value was calculated according to the two-tailed unpaired *t*-test or the one-way ANOVA with Tukey's multiple comparisons test, as specified for each experiment. All figures are representative of at least three biological replicates ± SEM, unless stated otherwise. For animal studies, no statistical methods were used to determine the sample size. When animals from the same cohort were used, these were randomly allocated in the different groups. Data obtained from animal experiments were blindly assessed. Results were indicated as outliers by GraphPad Prism software were removed from the analysis.

IHC and *in situ* hybridization and quantification, Western blot, Immunofluorescence staining, β-catenin luciferase activity assay, cancer cell adhesion to EC monolayer and transendothelial migration, and permeability assay protocols can be found in the Appendix Supplementary Methods.

## Data availability

The .raw MS files and search/identification files obtained with MaxQuant have been deposited to the ProteomeXchange Consortium (http://proteomecentral.proteomexchange.org/cgi/GetDataset) via the PRIDE partner repository (Vizcaino *et al*, 2013) with the dataset identifier PXD003316. Comparison proteins were regulated by high stiffness in HUVECs and YAP/TAZ gene targets identified by ChIP in MDA-MB-231 (Zanconato *et al*, 2015).

**Expanded View** for this article is available online.

## Acknowledgements

We would like to thank David Strachan for developing the ImageJ macro for analysis of tumor vascularity and pericyte coverage and Margaret O'Prey from the Beatson Advanced Imaging Resource, the BSU and histology facilities at the CRUK Beatson Institute, Evarest Onwubiko and Christopher Baxter for help with *in vivo* work, Clare Orange for histopathology services, NHS Greater Glasgow and Clyde Biorepository for the umbilical cords, and Prof Hans Jorg Fehling and the European Mouse Mutant Archive (EMMA) for providing the *Rosa26^flSTOP-tdRFP* mice. We thank Jim Norman for fruitful discussions, and Jon Lakins and Valerie Weaver for the improvements in the methodology of generating PAGs for cell culture. This work was funded by Cancer Research UK (CRUK Beatson Institute C596/A17196, CRUK Glasgow Centre C596/A18076 and S.Z. C596/A12935). We thank the PRIDE team.

## Author contributions

Conceptualization: SER, MM, LMC, and SZ; Methodology: SER, DMB, KB, MM, LMC, and SZ; Investigation: SER, LJN, JM, EJM, CN, EK, AS, FP, SD, A-TH, JS, AR-F, VP, KS, and SZ; Resources: YE, RHA, KB, DA, and JRH-F; Data curation, SER, LMC, MM, and SZ; Writing original draft: SER, SZ; Visualization: SER and SZ; Supervision: SI, DMB, LMM, MS-S, LMC, MM, and SZ.

## Conflict of interest

The authors declare that they have no conflict of interest.

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
