## [Review Process File · The EMBO Journal]

Manuscript EMBO-2016-94912

Tumour matrix stiffness promotes metastatic cancer cell interaction with the endothelium

Steven E Reid, Emily Kay, Lisa J Neilson, Anne-Theres Henze, Jens Serneels, Ewan J McGhee, Sandeep Dhayade, Colin Nixon, John Mackey, Alice Santi, Karthic Swaminathan, Dimitris Athineos, Vasileios Papalazarou, Francesca Patella, Ivar Rom-n-Fern-ndez, Yasmin ElMaghloob, Juan Ramon Hernandez-Fernaund, Ralf H Adams, Shehab Ismail, David M Bryant, Manuel Salmeron-Sanchez, Laura M Machesky, Leo M Carlin, Karen Blyth, Massimiliano Mazzone, Sara Zanivan

Corresponding author: Sara Zanivan, CRUK Beatson Institute

Review timeline:	Submission date:	13 June 2016
	Editorial Decision:	13 July 2016
	Revision received:	30 April 2017
	Editorial Decision:	29 May 2017
	Revision received:	08 June 2017
	Accepted:	09 June 2017

Editor: Andrea Leibfried

Transaction Report:

1st Editorial Decision

13 July 2016

Thank you for submitting your manuscript for consideration by the EMBO Journal. It has now been seen by four referees whose comments are shown below. Please note that referee #4 comments mainly on the proteomics part of your work.

As you will see, the referees appreciate your study. However, they also point out that further support for your conclusions is required.

Given the referees' positive recommendations, I would like to invite you to submit a revised version of the manuscript, addressing the comments of all four reviewers.

Importantly, the knock-out strategy needs to be supported by controls (referee #1 point 2; referee #2 point 1; referee #3 point 1) and further support for the mechano-regulation is needed (referee #1 point 4, referee #2 point 2, referee #3 point 3). More insight into the role of beta-catenin is also required (referee #1 point 6, referee #2 point 6) and the referees note some technical issues that need to be addressed.

I thus realize that the revision is quite demanding and I can extend the revision time to six months should that be helpful. I should add that it is EMBO Journal policy to allow only a single round of revision, and acceptance of your manuscript will therefore depend on the completeness of your responses in this revised version.

When preparing your letter of response to the referees' comments, please bear in mind that this will form part of the Review Process File, and will therefore be available online to the community. For more details on our Transparent Editorial Process, please visit our website: http://emboj.embopress.org/about#Transparent_Process

Thank you for the opportunity to consider your work for publication. I look forward to your revision.

REFEREE REPORTS

Referee #1:

In this work Reid et al study how changing matrix stiffness affects endothelial cells. Mass spectroscopy analysis reveals that endothelial cells on stiffer substrates have higher levels of CCN1/Cyr61; in addition, CDH2/N-cadherin is also up-regulated. Based on this the authors suggest that high levels CCN1/Cyr61 is linked to more productive CDH2-dependent cancer cell-endothelial cell interactions and increased metastasis. The subject is interesting and some of the ideas are new. While many studies have investigated links between tissue stiffness and cancer cell signaling, the analysis of how altered tissue stiffness affects endothelial cells is much less studied. Unfortunately there are currently several problems with the experiments and their interpretation. However, these problems are not unsurmountable. If the authors were able to robustly address them then the work could make a good contribution to EMBO J.

Specific comments

1. Figure 1f: blebbistatin treatment does not affect the induction of CCN1 by stiff matrix; instead it affects the basal level. This suggests that the 'mechano-response' is myosin independent.
2. The in vivo knock-out of CCN1/Cyr61 is simply not convincing. The authors use a strategy that relies on IV delivery of a cell permeable (TAT) Cre. However, there are many issues with this. The authors only show a non-significant two-fold increase in LacZ expression (which will occur after Cre-mediated recombination - Figure S2); this is not at all convincing. The authors should perform β -gal assays on tissue sections to demonstrate efficient endothelial specific recombination. In relation to this, the authors must analyze the extent of recombination in myeloid cells and stromal fibroblasts. Further, the authors should inject a non-functional but cell permeable Cre molecule as a control to exclude non-specific effects on the uptake of the TAT protein. In parallel, they should administer the functional Cre molecule to tumor bearing mice that lack the floxed allele. Finally, immunostaining for CCN1/Cyr61 on tumor sections must be shown.
3. How stiff are B16 tumors? 22kPa may be much stiffer than B16 tumors.
4. The authors should show in a second endothelial cell culture that CCN1/Cyr61 is mechano-regulated. The data with fibroblasts and cancer cells are rather peripheral; additional information with other endothelial cells will be important.
5. The data in Figure 2f & h are weak and probably not statistically significant. Why is MLANA mRNA found in fewer mice than have metastases? This suggests the PCR is not working properly. Further, Fig 2g is not significant.
6. The authors suggest that CDH2 is regulated by β -catenin. Does β -catenin ChIP on the CDH2 promoter. Also, why does CCN1-GFP still induce CDH2 when β -cat is depleted in Fig 3h? The basal CDH2 is affected, but its levels are still doubled by CCN1-GFP.
7. Can the authors show reduced CDH2 staining in the tumor sections that received the TAT-Cre?
8. Does β -catenin depletion in endothelial cells affect cancer cell interaction?
9. What about the role of CCN1/Cyr61 and CDH2 on in vitro trans-endothelial migration assays? The adhesion/intercalation assay is not sufficient, especially as cancer cells tend to get lodged in capillaries simply due to their large size.
10. Does CDH2 siRNA reduce PC3 and/or B16 metastasis following tail vein injection?
11. The authors should show the key CCN1 siRNA experiments with multiple independent siRNA.

Referee #2:

In their manuscript, the authors show that endothelial cells (ECs) respond to ECM stiffness modulation by increasing the production and secretion of matrix protein CCN1. CCN1 in turn activates b-catenin signaling to upregulate the expression of N-cadherin by ECs, thus enhancing the interaction between tumor cells and the endothelium, a critical step for extravasation and metastatic dissemination.

In general, using Mass-spec analysis to identify proteins regulated by matrix stiffness in ECs was comprehensive and well executed. The observation that CCN1, a YAP target gene, is induced in response to increasing stiffness, makes sense since YAP is known to be activated by matrix stiffness. The metastasis suppressing effect of knocking out CCN1 in ECs is very interesting and worth further investigation. However, it is unclear whether the described CCN1 knockout phenotype is indeed related to matrix stiffness. The follow-up mechanistic studies lack the depths needed to reveal the function of CCN1 in ECs to mediate its effect on metastasis.

Major points:

- 1- The authors use an elegant in vivo model to knockout the expression of CCN1 specifically in the endothelium by injecting a soluble form of Cre recombinase in the blood circulation. It is important to show by use immunofluorescent stainings the localization and expression changes of the mouse CCN1 in endothelium under the HTNC-treated condition and the control condition, just showing a 3-fold increase of lacZ mRNA in the lung is not sufficient. Moreover, given the extensive HTNC treatment after tumor implantation, it remains unclear whether CCN1 secreted by tumor cells could also contribute to this process. Indeed, FigS4J shows that knocking down CCN1 in the PC3 cells also significantly reduces tumor cells adhesion to ECs in vitro, suggesting that both tumor-derived and endothelium-derived CCN1 contribute importantly to the described process. Further in vivo investigation using CCN1 KD B16F10 cells in BSA and HTNC treated backgrounds is needed to clarify this issue.
- 2- Although Fig. 2 shows that CCN1 from the endothelial cells and/or tumor cells could impact metastasis, it is unclear whether this result relates to matrix stiffness-induced CCN1 function in ECs, which is the key novel conclusion that the authors would like to draw. Therefore, it is important to determine whether changing matrix stiffness in vivo could promote metastasis in a CCN1-dependent manner.
- 3- Fig 2H shows that EC-specific CCN1 KO significantly decreases the presence of CTCs compared to the control condition, supporting the hypothesis that the EC-derived CCN1 greatly contributes to the intravasation step. The authors could also perform tail vein injection of B16F10 cells in the BSA/HTNC-treated mice to test whether extravasation from endothelium is affected by EC-specific CCN1 KO.
- 4- Given the proposed role of CCN1 in ECs for intravasation into the circulation, the authors could use co-culture and tumor cell transendothelial migration assay to investigate in vitro the role of CCN1 in tumor cells-ECs interaction in addition to the cancer cell adhesion assay provided in Fig4.
- 5- ECM stiffness triggers CCN1 expression and secretion by ECs. CCN1 can in turn activate the beta-catenin signaling. Secreted CCN1 has been shown previously to signal through several integrins (alpha6beta1, alphaVbeta5, alphaVbeta3) and thus activate different processes (adhesion, migration, proliferation). The authors should test whether integrins or other receptors are involved in the regulation of b-catenin activity by CCN1. Is the ILK-GSK3b pathway (described in Xie et al, Cancer Research, 2004) involved?
- 6- Furthermore, overexpression of CCN1 only resulted in 2-fold increase in CDH2, which is dependent on beta-catenin, shown in Fig. 3H. It is important to determine whether under increasing matrix stiffness, whether beta-catenin is indeed required for CDH2 induction.

Minor points:

- 1- A panel showing the mRNA expression levels of CCN1 should be added in Fig1F.
- 2- In Fig4B (left panel), 400Pa should be changed to 1050Pa.
- 3- Fig4A (left panel) should be combined with FigS4A. the DAPI channel should be added.
- 4- Fig2C, D, E should be placed in supplementary FigS2.
- 5- The main conclusion of the manuscript is that CCN1 upregulation in ECs upon increasing matrix stiffness functions cell-autonomously on ECs to induce N-cadherin, thus promoting tumor cell adhesion and intravasation through ECs. The model presented in Fig. 4E seems to suggest that most

CCN1 functions on tumor cells?

Referee #3:

The manuscript entitled 'Tumour matrix stiffness promotes metastatic cancer cell interaction with the endothelium' by Reid et al. describes the identification of the matricellular protein CCN1 as a substrate rigidity dependent regulator of protein expression in endothelial cells (ECs) that impacts tumor metastasis through regulation of N-cadherin expression.

To investigate how protein expression is affected by substrate rigidity, the authors cultured ECs on matrices with distinct elastic moduli (400 Pa vs. 22 kPa) and used proteomics to identify regulated proteins. One of identified molecules is CCN1 that has been previously described to be regulated by the transcriptional co-activator YAP (Quan et al., 2014). As increased matrix stiffness, which is a hallmark of many solid tumors, has been shown to promote tumor growth and metastasis, the authors tested whether CCN1 expression is critical for these processes by combining EC-specific CCN1 depletion with an *in vivo* melanoma mouse model. Those experiments showed that reduction in CCN1 protein levels indeed reduce tumor metastasis.

To evaluate possible molecular mechanisms a range of cell culture experiments were performed which demonstrate that CCN1 expression affects protein expression of many molecules including N-cadherin. Additional experiments suggest that CCN1-dependent upregulation of N-cadherin expression requires β -catenin expression. Finally, the authors used a co-culture model to demonstrate that CCN1 and N-cadherin expression correlates with the ability of prostate cancer cells to adhere to underlying ECs. Together, these data are consistent with the hypothesis that a matrix-rigidity dependent increase in CCN1 expression facilitates cancer cell metastasis by upregulation of N-cadherin in endothelial cells.

Overall, the manuscript is well-written and figures have been carefully prepared. The experiments are sound, the data are, except from a few exceptions (see below), convincing and the reported differences seem statistically significant. The findings are very interesting and should be published; however, the manuscript would be further strengthened if a few control experiments were included; please see my suggestions below.

Major points:

1. The authors have used the HTNC system to induce the CCN1 knockout in the vasculature of mice but it is not really clear how efficient this strategy really is. The provided data convincingly demonstrate that CCN1 protein levels are reduced *ex vivo* (Fig. S2A) and LacZ mRNA becomes detectable in lung tissue of mice, but it seems possible that CCN1 depletion is incomplete. Could the absence of any phenotype regarding tumor growth and tumor vascularization also be explained by an incomplete knockout of CCN1? Thus, even though the HTNC system has been used before (Giacobini et al., 2014), I suggest to perform a LacZ staining of tissue sections to directly demonstrate the efficiency of this strategy.

By the same token, it is unclear how strongly CCN1 protein levels are reduced in the vasculature of mice. Is it not possible that CCN1 could be secreted by other cell types or remain associated in the extracellular matrix after knockout induction. The most convincing evidence for the lack of CCN1 would obviously be a CCN1 immunostainings of tissue sections to directly show the absence of CCN1. I am aware that this experiment depends on the availability of suitable antibodies, but it seems that such antibodies can be purchased (e.g. Anti-CCN1, abcam 24448).

2. The authors observed a significant downregulation of N-Cadherin in CCN1-depleted ECs *in vitro* (Fig. 3A). Can such a downregulation also be observed in tissues of CCN1 knockout mice? Again, this could be checked by immunostainings with available N-cadherin antibodies.

3. The authors seeded ECs on very soft (400 Pa) and on more rigid substrates (22 kPa) to perform the initial proteomics screen. However, the authors chose a stiffness of 1050 Pa in the co-culture system, because ECs apparently do not form confluent monolayers on 400 Pa matrices. Does this not imply that the chosen stiffness of 400 Pa was actually too soft to mimic a physiologically relevant context? I think that this is a potentially important issue and the authors should at least comment on it.

Minor points:

1. Fig.S3D and Fig.S4L should be replaced with blots of higher quality. It seems that the contrast was strongly adjusted.

2. In some western blots, CCN1 appears as single band (Fig.S3C or Fig.S4D) whereas other blots

show a double band (e.g. Fig.S3D or FigS4G). Why is that?

3. The provided data demonstrate that the expression of many molecules is affected by CCN1, for instance integrin alpha2 (Fig. 3A). It might be helpful if the authors would comment on potential effects through CCN1-regulated proteins other than N-cadherin in the discussion.

Referee #4:

In this manuscript Reid et al study the effect of substrate stiffness on protein expression in endothelial cells, aiming to understand how increased stiffness in the tumour matrix may impact tumour growth and metastasis. To investigate this, the authors use a cell line-based model system (HUVEC cells grown on different substrates) employing SILAC labeling for quantitative proteomics. Functionality of the secreted protein CCN1, one of the proteins that was differentially expressed, was tested *in vivo* by transplanting B16F10 melanoma cells in CCN1 proficient or deficient mice. This showed that CCN1 contributed to metastasis to the lung in an N-cadherin-dependent manner.

The influence of matrix stiffness (or other properties of the tumor environment) on protein expression cannot be easily assessed *in vivo*, and therefore the authors resort to a rather artificial *in vitro* system using a normal endothelial cells line (HUVEC) grown on polyacrylamide gels of different stiffness. The validity of the system is nicely shown in Fig 1a/b. The authors then follow a sound and tested SILAC labeling protocol to examine protein expression, in combination with a clever experiment to correct for effects in protein expression due to differences in proliferation rather than stiffness *per se* (Fig 1d).

Comments:

1. In the performed SILAC experiment, the authors use rather relaxed cutoff criteria to call differentially expressed proteins (>1 SD from the mean, while usually 2 or 2.5 SD is used) (Fig S1c). In addition, it is not clear what criteria were used to distinguish changes in protein expression that were contributed to stiffness and proliferation, resp (Fig 1E). The figure legends seem to indicate that proteins were required to be regulated in not more than 1 (out of 2) replicate experiments - which is not very strict. Indeed the heat map (left and right panels in Fig 1E) show rather heterogeneous patterns and a considerable number of missing data/ratios especially for the proliferation experiments. Although the authors go on to demonstrate the biological role of CCN1, it is difficult to estimate what fraction of the other proteins mentioned in the figure are deemed to be functional due to the lack of (replicate) data in one or more of the experimental conditions. The authors should more explicitly describe how they treated the data in the methods section rather than in a single sentence in the figure legends.
2. Why were different procedures used for proteomic sample preparation (in gel and in solution digestion, page 13)? This might introduce a bias e.g. because proteins elute/digest differently, or because of differences in obtained proteome coverage.
3. The authors investigate CCN1 to assess its stiffness-induced expression, however they do not comment on any of the other 'stiffness' proteins (Fig 1E). Is there any pattern/consistency in their known functionality? Or can any of them, with the hindsight of the role and mechanism of CCN1, be placed in the context of the proposed model (Fig 4E)?

Point by point response to reviewers

Summary:

We thank all the reviewers for the critical evaluation of our manuscript. The comments that they have provided have been extremely helpful to prepare a revised version of the manuscript, which we believe is now clearer and more comprehensive.

Based on the reviewers' comments, most of the Figures have been made new and additional data have been integrated into the old Figures. In particular Figure 1, Figure 2, Figure 4, Figure 5, Figure 6 and Figure 7, Extended View Figure 3, Appendix Figure S1, Appendix Figure S4, Appendix Figure S5, Appendix Figure S6, Movie EV1 and Movie EV2 contain new data. We have also represented differently the Proteomic data in Extended View Figure 1 and Figure 3.

The most important addition to the revised manuscript is the in vivo work, which was a concern for Reviewers #1, #2 and #3. Using a Cre inducible RFP reporter mice model (*Rosa26^{fSTOP-tdRFP}*) we show that HTNC i.v. treatment is capable, to some extent, to induce recombination in the tumour vessels (Appendix Figure S4). Moreover, we provide conclusive prove that endothelial CCN1 controls endothelial cell-cancer cell binding in vivo. We have generated inducible endothelial specific (*Cdh5-Cre^{ERT2}*) Ccn1 knock out mice (*Ccn1 KO^{EC}*) to deplete Ccn1 in the endothelium of adult mice, and performed intravital imaging analysis of fluorescently labelled B16F10 melanoma cells intradermally transplanted in *Ccn1 KO^{EC}* mice or *Ccn1 WT*, as control. We show that cancer cells can stably or transiently bind to blood vessels and that the number of cancer cells that stably adhered to the blood vessels is significantly reduced upon depletion of Ccn1 in the endothelium (Figure 6A,B, Movie EV1 and Movie EV2).

As requested by Reviewer #1, we have corroborated the mechano-regulation of CCN1 observed in HUVECs also in human dermal microvascular endothelial cells (HMVECs) (Appendix Figure S1). We also show that CCN1 in HMVECs controls N-Cadherin levels and binding to cancer cells (Figure 5D and Appendix Figure S3G). Moreover, since this was a concern for Reviewer #3, we included references which clarify that 400 Pa is physiologically relevant stiffness for the endothelium in the Discussion.

Regarding the link between CCN1, β -catenin and N-Cadherin, to answer the concern of Reviewer #1, we have measured the transcriptional regulation of N-Cadherin dependent on β -catenin and CCN1 levels. This clearly shows that silencing β -catenin abrogates almost completely the increase in expression on N-Cadherin induced by CCN1-GFP over-expression (Figure 4H). To answer Reviewer #2 we show that β -catenin is required for stiffness induced levels of N-Cadherin. In fact when we silence β -catenin in HUVECs and plate these cells on polyacrylamide gels of low and high stiffness, while the levels of N-Cadherin are higher at high compared to low stiffness in the siCtl cells, the levels of N-Cadherin are similar at high and low stiffness in cells silenced for β -catenin (Appendix Figure S2A).

We have also provided a different representation of the proteomic data for a clearer discussion of the proteomic results (Figure EV1), and included more details about the analysis of the proteomic data in the Material Methods, as requested by Reviewer #4.

We would like to thank once more all the reviewers for their guidance to improve our manuscript.

Referee #1:

In this work Reid et al study how changing matrix stiffness affects endothelial cells. Mass spectroscopy analysis reveals that endothelial cells on stiffer substrates have higher levels of CCN1/Cyr61; in addition, CDH2/N-cadherin is also up-regulated. Based on this the authors suggest that high levels CCN1/Cyr61 is linked to more productive CDH2-dependent cancer cell-endothelial cell interactions and increased metastasis. The subject is interesting and some of the ideas are new. While many studies have investigated links between tissue stiffness and cancer cell signaling, the analysis of how altered tissue stiffness affects endothelial cells is much less studied. Unfortunately there are currently several problems with the experiments and their interpretation. However, these problems are not unsurmountable. If the authors were able to robustly address them then the work could make a good contribution to EMBO J.

Specific comments

1. Figure 1f: blebbistatin treatment does not affect the induction of CCN1 by stiff matrix; instead it affects the basal level. This suggests that the 'mechano-response' is myosin independent.

It is true that blocking myosin II resulted in reduced levels of CCN1 at any stiffness, and that it is still possible to measure a response to stiffness, to some extent. However, the increase in CCN1 levels between low and high (22KPa) stiffness was also reduced (CCN1 levels increased 5.5 fold between 400Pa and 22KPa in control cells, while 3 fold only upon Blebbistatin treatment). Moreover, we have now performed also RT-PCR on HUVECs cultured on PAGs and treated with Blebbistatin and this clearly shows that the response to stiffness is completely ablated by Blebbistatin treatment (Figure 1H,I). The discrepancy in results (protein and mRNA levels) is likely due to the fact that CCN1 protein can be accumulated over time in the matrix produced by the cells in culture.

2. The in vivo knock-out of CCN1/Cyr61 is simply not convincing. The authors use a strategy that relies on IV delivery of a cell permeable (TAT) Cre. However, there are many issues with this. The authors only show a non-significant two-fold increase in LacZ expression (which will occur after Cre-mediated recombination - Figure S2); this is not at all convincing. The authors should perform β -gal assays on tissue sections to demonstrate efficient endothelial specific recombination. In relation to this, the authors must analyze the extent of recombination in myeloid cells and stromal fibroblasts. Further, the authors should inject a non-functional but cell permeable Cre molecule as a control to exclude non-specific effects on the uptake of the TAT protein. In parallel, they should administer the functional Cre molecule to tumor bearing mice that lack the floxed allele. Finally, immunostaining for CCN1/Cyr61 on tumor sections must be shown.

We agree with the reviewer that more validation of the method used to deplete Ccn1 in the blood vessels would strengthen the results. We tried several Lac-Z stainings on the tissues that were left over from that experiment, but none of them was successful. This could be due to the fact that Lac-Z staining should be performed on freshly isolated tissues. We have removed the LacZ expression data the reviewer refers to and instead decided to evaluate the extent of recombination obtained through HTNC i.v. administration using a Cre inducible reporter mouse model *Rosa26^{f1STOP-tdRFP}*. We

subcutaneously transplanted B16F10 melanoma cells in *Rosa26^{f1STOP-tdRFP}* mice treated with HTNC or BSA, as a control. We observed that recombination occurred within the vasculature in some regions of the tumour with varying degree of efficiency (estimated to be 0-75% of the total tumour vasculature, depending on the region) (Appendix Figure S4B,C). In the lung, we observed that the HTNC induced recombination in some cells in the lungs (estimated to be 0-2% of the total lung vasculature, depending on the regions). Following the reviewer's suggestion of analysing the recombination in the myeloid cells, we used flow cytometry analysis to assess that the circulating HTNC did not affect myeloid cells. In fact, we could not detect RFP⁺ staining in myeloid cells isolated from blood or bone marrow. Regarding the effects of HTNC in the tumour stroma, we found that B16F10 tumours have only small stromal regions and we could not detect RFP⁺ staining in fibroblast-like cells. These results prove that HTNC i.v. treatment is capable, to some extent, to induce recombination in the tumour vessels.

Unfortunately we cannot provide any reliable staining of *Ccn1* in *Ccn1^{fl/fl}* mice treated with HTNC because we could not find any antibody that works well in IHC/IF on tissue samples. We have now RNAscope probe for in situ hybridization that can be used for this purpose, but unfortunately the fixing of the tissues obtained from the previous experiment was not compatible with this staining. We have stained for *Ccn1* (in situ hybridization) B16F10 tumours that grew in the *Rosa26^{f1STOP-tdRFP}* mice and this showed that tumour vessels express high levels of *Ccn1*, thus highlighting the relevance of the B16F10 tumour model to study vascular *Ccn1*. These results are included in Appendix Figure S4.

We understand the concerns of the reviewer about the use of the HTNC protein, and although the negative control suggested would provide further evidence that the Cre is functional and specific, we instead hope that using a different setup, by generating inducible endothelial specific (*Cdh5-Cre^{ERT2}*) *Ccn1* knock out mice, supports the findings from the HTNC experiments that the observations are indeed from reduction of *Ccn1* and not a non-specific effect of the TAT protein. We have generated inducible endothelial specific *Ccn1* knock out mice (*Ccn1 KO^{EC}*) and show that *Ccn1* is expressed in some regions of the lung vasculature of adult mice and that, upon induction with tamoxifen, *Ccn1* expression (RNAscope in situ hybridization) is completely lost in the lungs and that this does not influence total lung vascularity (Figure EV3B-F). We have then used intravital imaging analysis and provide clear evidence that endothelial CCN1 control endothelial-cancer cell binding in vivo. We have transplanted fluorescently labelled B16F10 melanoma cells intradermally in the ear of *Ccn1 KO^{EC}* and *Ccn1 WT* mice and used intravital imaging to monitor the binding of the cancer cells to fluorescently labelled blood vessels. This showed that, as in our *in vivo* set up, when endothelial cells lack *Ccn1*, cancer cells adhere less to the blood vessels (Figure 6A,B). We hope that the additional in vivo experiments we have added satisfy the reviewers concerns on the role of CCN1 in the metastasis cascade.

3. How stiff are B16 tumors? 22kPa may be much stiffer than B16 tumors.

We could not measure the stiffness of B16F10 tumours. However, we could perform staining for *Ccn1* and show that its expression in the vessels of B16F10 tumours is very high (RNAscope in situ hybridization, Appendix Figure S4E), thus highlighting the relevance of the B16F10 model to study the function of endothelial CCN1 in the tumour context.

In addition, we show that there can be a relationship between tumour stiffness and Ccn1 expression using E0771 orthotopic tumours. Using Sirius Red staining to visualise collagen fibres, RNAscope in situ hybridization to evaluate Ccn1 expression and Atomic Force Microscopy to measure tumour stiffness on tumour sections, we show that high Ccn1 is found in tumour regions with high collagen content and that high collagen content is found in regions of the tumours with high stiffness (Figure 2). Unfortunately we could not use this model for further experiments because RNAscope in situ hybridization for N-Cadherin showed that, while E0771 cells express N-Cadherin in vitro, this is not the case when cells have been grown orthotopically in mice for two weeks.

4. The authors should show in a second endothelial cell culture that CCN1/Cyr61 is mechano-regulated. The data with fibroblasts and cancer cells are rather peripheral; additional information with other endothelial cells will be important.

We have now included data showing that also human dermal microvascular endothelial cells (HMVEC) increase levels of CCN1 with increasing stiffness and that this is mechano-regulated (Appendix Figure S1B). We also show that CCN1 in HMVEC controls N-Cadherin levels and binding to cancer cells (Figure 5D and Appendix Figure S3G).

5. The data in Figure 2f & h are weak and probably not statistically significant. Why is MLANA mRNA found in fewer mice than have metastases? This suggests the PCR is not working properly. Further, Fig 2g is not significant.

In the experiment that we have performed, the number of metastasis in the lungs was rather low (sometimes only one). We believe that for this reason we haven't been able to detect MLANA mRNA in all the mice with lung metastasis. We have removed Figure 2G and now present only the incidence of metastases and CTCs, which shows the proportion of mice with detected metastasis or CTCs in the CTL and HTNC treated groups, indicating that HTNC treated mice are less likely to have CTCs present or form metastases.

6. The authors suggest that CDH2 is regulated by β -catenin. Does β -catenin ChIP on the CDH2 promoter. Also, why does CCN1-GFP still induce CDH2 when β -cat is depleted in Fig 3h? The basal CDH2 is affected, but its levels are still doubled by CCN1-GFP.

We thank the Reviewer for this comment. In fact, we did not mean to suggest that β -catenin directly regulates N-Cadherin expression, as we do not have data supporting this. Our data indicate though that CCN1-dependent N-Cadherin expression requires β -catenin. However, likely, there is another transcription factor involved between β -catenin and N-Cadherin. Supporting this hypothesis, we have treated HUVECs cultured at high stiffness (culture dish) with cyclohexamide to block translation and measured the expression levels of N-Cadherin. If β -catenin directly controls N-Cadherin expression we would expect the levels of N-Cadherin not to change. However, after 30 mins treatment with cyclohexamide, N-Cadherin expression was strongly downregulated (Reviewer Figure 1). Thus, these data suggest that there might be other transcription factors whose translation is

required to control N-Cadherin levels in HUVECs. For this reason, we haven't performed a ChIP assay.

Reviewer Figure 1. HUVECs cultured on plastic dishes were treated with Cyclohexamide for the indicated time to inhibit protein translation. N-Cadherin (CDH2) levels have been normalised by the levels of the housekeeping gene S18 Ribosomal Subunit (similar results obtained normalising by GAPDH or TBP housekeeping genes).

We agree with the Reviewer that the western blot (Figure 4G in the revised manuscript) shows that, in cells silenced for β -catenin, the overexpression of CCN1-GFP increases the levels of N-Cadherin. This can be explained with the fact that there was not a complete silencing of β -catenin and that the over-expression of CCN1-GFP induced an increase in β -catenin also in the β -catenin silenced cells (as shown in the western blot in Figure 4G). To investigate this further, we have measured also the transcriptional regulation of N-Cadherin dependent on β -catenin and CCN1 levels. This clearly shows that silencing β -catenin almost abrogates the increase in expression on N-Cadherin induced by CCN1-GFP over-expression (Figure 4H).

7. Can the authors show reduced CDH2 staining in the tumor sections that received the TAT-Cre?

Unfortunately we could not find any antibody able to give a good/reliable staining of N-Cadherin on those FFPE and frozen tissues. However, we were able to show that Ccn1-endothelial depleted mice (*Ccn1* KO^{EC}) have lower levels of N-Cadherin in the lungs by using RNAscope in situ hybridization (Figure EV3G).

8. Does β -catenin depletion in endothelial cells affect cancer cell interaction?

Yes, it does. We have now performed an adhesion assay of PC3 cells on HUVECs silenced for β -catenin and show that there is a 75% inhibition of the binding when HUVECs are silenced for β -catenin, compared with their control counterpart (Figure 5E).

9. What about the role of CCN1/Cyr61 and CDH2 on in vitro trans-endothelial migration assays? The adhesion/intercalation assay is not sufficient, especially as cancer cells tend to get lodged in capillaries simply due to their large size.

We have tried many trans-endothelial migration protocols (and many combination of endothelial cell and cancer cell types), and found that generally cancer cells migrate between the membrane of the transwell and the endothelial monolayer, without actually going through it, as reported in many works. We have found however that the highly invasive PC3 clone TEM4-18 (Drake, J.M. et al. Mol Biol Cell 20, 2207-2217 (2009)) can protrude through HMVEC cells at some extent and that silencing CCN1 in the HMVEC reduces transendothelial migration of those cancer cells (Figure 7C,D). To corroborate these data, we have also used an alternative approach. We have measured the effects of letting PC3 cells to adhere for 24h, rather than 40 mins, on an HUVEC monolayer. This assay clearly showed that while after 40 mins the cancer cells adhere to the endothelial cells without affecting the integrity of the monolayer, after 24h the cancer cells create holes within the monolayer. Quantification of the monolayer disruption clearly showed that when CCN1 is silenced in HUVEC, the monolayer keeps a higher level of integrity (Figure 7A,B), thus indicating that in the absence of CCN1, cancer cells are less likely to break the endothelial barrier.

10. Does CDH2 siRNA reduce PC3 and/or B16 metastasis following tail vein injection?

This would indeed be an interesting question to answer. However, we did not perform the suggested experiment because we could not find differences in B16F10 lung metastasis following tail vein injection in endothelial Ccn1 depleted mice (*Ccn1*^{KO^{EC}) (Appendix Figure S5G-I). We think that this could be due to the fact that only low levels of N-Cadherin can be detected in normal adult lungs (Figure EV3G). However, it is not known whether N-Cadherin levels in the blood vessels are altered in the pre-metastatic niche. If that would be the case, adequate mouse models should be used to investigate the role of cancer cell-N-Cadherin in metastasis. We thought that answering this question was therefore beyond the aim of this manuscript, but we addressed this point in the Discussion.}

11. The authors should show the key CCN1 siRNA experiments with multiple independent siRNA.

We have now performed key experiments (N-Cadherin regulation and EC-cancer cell binding) with independent siRNA. This showed results comparable to those obtained with the siRNA pool for CCN1 (Figure 4A; Figure EV2F and Figure 5C).

Referee #2:

In their manuscript, the authors show that endothelial cells (ECs) respond to ECM stiffness modulation by increasing the production and secretion of matrix protein CCN1. CCN1 in turn activates b-catenin signaling to upregulate the expression of N-cadherin by ECs, thus enhancing

the interaction between tumor cells and the endothelium, a critical step for extravasation and metastatic dissemination.

In general, using Mass-spec analysis to identify proteins regulated by matrix stiffness in ECs was comprehensive and well executed. The observation that CCN1, a YAP target gene, is induced in response to increasing stiffness, makes sense since YAP is known to be activated by matrix stiffness. The metastasis suppressing effect of knocking out CCN1 in ECs is very interesting and worth further investigation. However, it is unclear whether the described CCN1 knockout phenotype is indeed related to matrix stiffness. The follow-up mechanistic studies lack the depths needed to reveal the function of CCN1 in ECs to mediate its effect on metastasis.

Major points:

1- The authors use an elegant *in vivo* model to knockout the expression of CCN1 specifically in the endothelium by injecting a soluble form of Cre recombinase in the blood circulation. It is important to show by use immunofluorescent stainings the localization and expression changes of the mouse CCN1 in endothelium under the HTNC-treated condition and the control condition, just showing a 3-fold increase of lacZ mRNA in the lung is not sufficient. Moreover, given the extensive HTNC treatment after tumor implantation, it remains unclear whether CCN1 secreted by tumor cells could also contribute to this process. Indeed, FigS4J shows that knocking down CCN1 in the PC3 cells also significantly reduces tumor cells adhesion to ECs *in vitro*, suggesting that both tumor-derived and endothelium-derived CCN1 contribute importantly to the described process. Further *in vivo* investigation using CCN1 KD B16F10 cells in BSA and HTNC treated backgrounds is needed to clarify this issue.

We agree with the reviewer that more validation of the method used to deplete *Ccn1* in the blood vessels would strengthen the results. We tried several Lac-Z stainings on the tissues that were left over from that experiment, but none of them was successful. This could be due to the fact that Lac-Z staining should be performed on freshly isolated tissues. We have removed the LacZ expression data the reviewer refers to and instead decided to evaluate the extent of recombination obtained through HTNC *i.v.* administration using a Cre inducible reporter mouse model *Rosa26^{f1STOP-tdRFP}*. We subcutaneously transplanted B16F10 melanoma cells in *Rosa26^{f1STOP-tdRFP}* mice treated with HTNC or BSA, as a control. We observed that recombination occurred within the vasculature in some regions of the tumour with varying degree of efficiency (estimated to be 0-75% of the total tumour vasculature, depending on the region) (Appendix Figure S4B,C). In the lung, we observed that the HTNC induced recombination in some cells in the lungs (estimated to be 0-2% of the total lung vasculature, depending on the regions). These results prove that HTNC *i.v.* treatment is capable, to some extent, to induce recombination in the tumour vessels. To strengthen the previous results, and further confirm our *in vitro* data, we have generated inducible endothelial specific (*Cdh5-Cre^{ERT2}*) *Ccn1* knock out mice (*Ccn1 KO^{EC}*). We show that *Ccn1* is expressed in some regions of the lung vasculature of adult mice and that, upon induction with tamoxifen, *Ccn1* expression (RNAscope *in situ* hybridization) is completely lost in the lungs and that this does not influence total lung vascularity (Figure EV3B-F). We have then used intravital imaging analysis and provide clear evidence that endothelial CCN1 control endothelial-cancer cell binding *in vivo*. We have transplanted fluorescently labelled B16F10 melanoma cells intradermally in the ear of *Ccn1 KO^{EC}* and *Ccn1 WT*

mice and used intravital imaging to monitor the binding of the cancer cells to fluorescently labelled blood vessels. This showed that, as in our *in vivo* set up, when endothelial cells lack *Ccn1*, cancer cells adhere less to the blood vessels (Figure 6A,B). We hope that the additional *in vivo* experiments we have added satisfy the reviewers concerns on the role of CCN1 in the metastasis cascade.

Regarding the impact of HTNC on cancer cell-derived CCN1, HTNC treatment should not have any impact on the cancer cells, because the B16F10 cells are not *Ccn1^{fl/fl}*. For this reason, we did not perform any experiments using *Ccn1* KD B16F10 cells. Moreover, it has recently been shown that, in MDA-MB-231 breast cancer cells, CCN1 expression enhances lung metastasis by promoting extravasation of the cancer cells from the blood circulation and inhibiting anoikis (Huang, Y.T. et al. Oncotarget 8, 9200-9215 (2017)). For this reason we believe that it would be quite difficult to interpret the results of the proposed experiment.

Regarding the results in Figure S4J and Figure 4D in the original version of the manuscript, those experiments show that silencing CCN1 in the cancer cells reduced N-Cadherin levels. These reduced levels of N-Cadherin in the cancer cells are the reason why siCCN1 cancer cells do not bind as much as the control cells to the endothelial monolayer. In Figure 4C (Figure 5F in the revised manuscript), in fact, we used an antibody that blocks N-Cadherin to show that the binding between cancer cells and endothelial cells depends on N-Cadherin.

2- Although Fig. 2 shows that CCN1 from the endothelial cells and/or tumor cells could impact metastasis, it is unclear whether this result relates to matrix stiffness-induced CCN1 function in ECs, which is the key novel conclusion that the authors would like to draw. Therefore, it is important to determine whether changing matrix stiffness *in vivo* could promote metastasis in a CCN1-dependent manner.

We agree with this Reviewer that with the *in vivo* experiments performed we cannot prove that the effects that we have observed upon depletion of *Ccn1* in the endothelial cells are stiffness dependent. We thank the reviewer for the suggested experiments, however, manipulating the stiffness of the entire tumour would have strong impact on the behaviour not only of the endothelial cells, but also cancer cells and other stromal cells, since it has been shown that blocking LOX induced collagen crosslinking results in reduced tissue stiffness, tumour incidence, metastasis, and delays cancer progression (Miller et al., 2015, Levental et al., 2009). The results would therefore be difficult to interpret in the context of *Ccn1* depletion in the vasculature. To answer to the Reviewer's concern we have been careful to avoid such a statement when interpreting the *in vivo* data. What we can clearly state though is that high/tumour matrix stiffness can induce strong alteration in the endothelial phenotype and that targeting such alterations in the tumour context can reduce metastasis. To strengthen previous data, we have now used different cancer and endothelial cell types to show that endothelial *Ccn1* promote the binding to cancer cells *in vitro* and used intravital imaging (B16F10 intradermal injection in the ear of endothelial specific conditional *Ccn1* *KO^{EC}* or *Ccn1* *WT* mice) to prove that this occurs also *in vivo* (Figure 6A,B).

3- Fig 2H shows that EC-specific CCN1 KO significantly decreases the presence of CTCs compared to the control condition, supporting the hypothesis that the EC-derived CCN1 greatly contributes to the intravasation step. The authors could also perform tail vein injection of B16F10 cells in the BSA/HTNC-treated mice to test whether extravasation from endothelium is affected by EC-specific CCN1 KO.

We have done tail vein injection of B16F10 cells using endothelial specific *Ccn1* KO mice (*Cdh5-Cre^{ER2}* conditional KO, *Ccn1* KO^{EC}) and could not see differences in lung metastasis (Appendix Figure S5G-I), thus indicating that the effect observed in lung metastasis is likely due to intravasation defects.

4- Given the proposed role of CCN1 in ECs for intravasation into the circulation, the authors could use co-culture and tumor cell transendothelial migration assay to investigate in vitro the role of CCN1 in tumor cells-ECs interaction in addition to the cancer cell adhesion assay provided in Fig4.

We have tried many trans-endothelial migration protocols (and many combinations of endothelial cell and cancer cell types), and found that generally cancer cells migrate between the membrane of the transwell and the endothelial monolayer, without actually going through it. We have found however that the highly invasive PC3 clone TEM4-18 (Drake, J.M. et al. Mol Biol Cell 20, 2207-2217 (2009)) can protrude through HMVEC cells at some extent and that silencing CCN1 in the HMVEC reduces transendothelial migration of those cancer cells (Figure 7C,D). To corroborate these data, we have also used an alternative approach. We have measured the effects of letting PC3 cells to adhere for 24h, rather than 40 mins, on an HUVEC monolayer. This assay clearly showed that while after 40 mins the cancer cells adhere to the endothelial cells without affecting the integrity of the monolayer, after 24h the cancer cells create holes within the monolayer. Quantification of the monolayer disruption clearly showed that when CCN1 is silenced in HUVEC, the monolayer keeps a higher level of integrity (Figure 7A,B), thus indicating that in the absence of CCN1, cancer cells are less likely to break the endothelial barrier.

5- ECM stiffness triggers CCN1 expression and secretion by ECs. CCN1 can in turn activate the beta-catenin signaling. Secreted CCN1 has been shown previously to signal through several integrins (alpha6beta1, alphaVbeta5, alphaVbeta3) and thus activate different processes (adhesion, migration, proliferation). The authors should test whether integrins or other receptors are involved in the regulation of b-catenin activity by CCN1. Is the ILK-GSK3b pathway (described in Xie et al, Cancer Research, 2004) involved?

(Unpublished data not included in the Peer Review Process File)

6- Furthermore, overexpression of CCN1 only resulted in 2-fold increase in CDH2, which is dependent on beta-catenin, shown in Fig. 3H. It is important to determine whether under increasing matrix stiffness, whether beta-catenin is indeed required for CDH2 induction.

We have now determined that β -catenin is required for stiffness induced levels of N-Cadherin. In fact when we silenced β -catenin in HUVECs and plated these cells on polyacrylamide gels of low and high stiffness, while the levels of N-Cadherin were higher at high compared to low stiffness in the siCtl cells, the levels of N-Cadherin were similar at high and low stiffness in cells silenced for β -catenin (Appendix Figure S2A).

Minor points:

1- A panel showing the mRNA expression levels of CCN1 should be added in Fig1F.

This has been included as Figure 1I.

2- In Fig4B (left panel), 400Pa should be changed to 1050Pa.

This has been changed (Figure 5B).

3- Fig4A (left panel) should be combined with FigS4A. the DAPI channel should be added.

This has been changed (Figure 5A and Appendix Figure S3A).

4- Fig2C, D, E should be placed in supplementary FigS2.

These Figures have been moved in the Appendix in the Revised Manuscript (Appendix Figure S5).

5- The main conclusion of the manuscript is that CCN1 upregulation in ECs upon increasing matrix stiffness functions cell-autonomously on ECs to induce N-cadherin, thus promoting tumor cell adhesion and intravasation through ECs. The model presented in Fig. 4E seems to suggest that most CCN1 functions on tumor cells?

We have made a new model that better represents our findings (Figure 7E).

Referee #3:

The manuscript entitled 'Tumour matrix stiffness promotes metastatic cancer cell interaction with the endothelium' by Reid et al. describes the identification of the matricellular protein CCN1 as a substrate rigidity dependent regulator of protein expression in endothelial cells (ECs) that impacts tumor metastasis through regulation of N-cadherin expression.

To investigate how protein expression is affected by substrate rigidity, the authors cultured ECs on matrices with distinct elastic moduli (400 Pa vs. 22 kPa) and used proteomics to identify regulated proteins. One of identified molecules is CCN1 that has been previously described to be regulated by the transcriptional co-activator YAP (Quan et al., 2014). As increased matrix stiffness, which is a hallmark of many solid tumors, has been shown to promote tumor growth and metastasis, the authors tested whether CCN1 expression is critical for these processes by combining EC-specific CCN1 depletion with an in vivo melanoma mouse model. Those experiments showed that reduction in CCN1 protein levels indeed reduce tumor metastasis.

To evaluate possible molecular mechanisms a range of cell culture experiments were performed which demonstrate that CCN1 expression affects protein expression of many molecules including N-cadherin. Additional experiments suggest that CCN1-dependent upregulation of N-cadherin expression requires β -catenin expression. Finally, the authors used a co-culture model to demonstrate that CCN1 and N-cadherin expression correlates with the ability of prostate cancer cells to adhere to underlying ECs. Together, these data are consistent with the hypothesis that a matrix-rigidity dependent increase in CCN1 expression facilitates cancer cell metastasis by upregulation of N-cadherin in endothelial cells.

Overall, the manuscript is well-written and figures have been carefully prepared. The experiments are sound, the data are, except from a few exceptions (see below), convincing and the reported differences seem statistically significant. The findings are very interesting and should be published; however, the manuscript would be further strengthened if a few control experiments were

included; please see my suggestions below.

Major points:

1. The authors have used the HTNC system to induce the CCN1 knockout in the vasculature of mice but it is not really clear how efficient this strategy really is. The provided data convincingly demonstrate that CCN1 protein levels are reduced ex vivo (Fig. S2A) and LacZ mRNA becomes detectable in lung tissue of mice, but it seems possible that CCN1 depletion is incomplete. Could the absence of any phenotype regarding tumor growth and tumor vascularization also be explained by an incomplete knockout of CCN1? Thus, even though the HTNC system has been used before (Giacobini et al., 2014), I suggest to perform a LacZ staining of tissue sections to directly demonstrate the efficiency of this strategy.

By the same token, it is unclear how strongly CCN1 protein levels are reduced in the vasculature of mice. Is it not possible that CCN1 could be secreted by other cell types or remain associated in the extracellular matrix after knockout induction. The most convincing evidence for the lack of CCN1 would obviously be a CCN1 immunostainings of tissue sections to directly show the absence of CCN1. I am aware that this experiment depends on the availability of suitable antibodies, but it seems that such antibodies can be purchased (e.g. Anti-CCN1, abcam 24448).

(Unpublished data not included in the Peer Review Process File)

2. The authors observed a significant downregulation of N-Cadherin in CCN1-depleted ECs in vitro (Fig. 3A). Can such a downregulation also be observed in tissues of CCN1 knockout mice? Again, this could be checked by immunostainings with available N-cadherin antibodies.

We now show that lower expression of N-Cadherin (RNAscope in situ hybridization) was found in the lungs of endothelial specific Ccn1 KO mice (*Ccn1* KO^{EC} , Figure EV3G). Importantly, upon endothelial specific depletion of Ccn1, almost all detectable Ccn1 expression in the lungs was lost (Figure EV3B,C), indicating that ECs are the primary source of Ccn1 in the lungs and thus the reduction of N-Cadherin staining reflect its regulation in the endothelial cells.

3. The authors seeded ECs on very soft (400 Pa) and on more rigid substrates (22 kPa) to perform the initial proteomics screen. However, the authors chose a stiffness of 1050 Pa in the co-culture system, because ECs apparently do not form confluent monolayers on 400 Pa matrices. Does this not imply that the chosen stiffness of 400 Pa was actually too soft to mimic a physiologically relevant context? I think that this is a potentially important issue and the authors should at least comment on it.

As it has been previously reported, 400Pa or even softer is physiologically relevant (e.g. Acerbi I. et al. *Integr Biol (Camb)*. 2015 Oct;7(10):1120-34). It has been observed that below 1000 Pa ECs form tubules on Matrigel and cells prefer cell-cell adhesion, but tubule formation stops above 1000 Pa, where cells are able to form greater substrate adhesions (Saunders et al. *Cellular and Molecular Bioengineering* 3, 60-67 (2010)). This implies that 400 Pa is relevant as physiological stiffness, however, it is not suitable for some in vitro experiments. We have now included the following two sentences and references in the manuscript:

Results:

“To simulate the initial process of cancer cell intravasation, we measured the adhesion of cancer cells to a confluent monolayer of ECs, where the lowest stiffness was replaced with 1050 Pa. At this stiffness cells are able to form greater substrate adhesions (Saunders & Hammer, 2010) and they formed an intact monolayer, while at 400 Pa we could not observe the formation of an intact monolayer.”

Discussion:

“We have explored the effect that the tumor matrix stiffness may have on the resident and attracted ECs within the tumor environment. For this we have used a range of physiological and pathological stiffness which have been shown to influence the endothelial phenotype. While at low, physiological stiffness, 400 Pa, endothelial cells form capillary-like networks in 2D and tubules in 3D matrix, at increasing stiffness endothelial cells assemble into networks with larger lumens and less branched (Saunders & Hammer, 2010; Sieminski et al, 2004).”

Minor points:

1. Fig.S3D and Fig.S4L should be replaced with blots of higher quality. It seems that the contrast was strongly adjusted.

We believe the reviewer means to highlight Figure S4M, since S4L has not been adjusted, and seems of rather good quality. This has now been replaced with a better quality western blot (Appendix Figure S3L).

Figure S3D is no longer in the manuscript since we have now used more extensively single siRNAs to validate siRNA pool for CCN1 (Figure 4A; Figure EV2F and Figure 5C).

2. In some western blots, CCN1 appears as single band (Fig.S3C or Fig.S4D) whereas other blots show a double band (e.g. Fig.S3D or Fig.S4G). Why is that?

It has been shown that CCN1 can carry post-translational modifications and that it has alternatively spliced transcripts, which have been detected at the mRNA level (J Cell Commun Signal. 2009 Jun; 3(2): 153–157). This could explain the presence of multiple bands for CCN1 by western blot. The fact that multiple bands could be found only in some blots may depend of the separation achieved by SDS-Page gel in each experiment.

3. The provided data demonstrate that the expression of many molecules is affected by CCN1, for instance integrin alpha2 (Fig. 3A). It might be helpful if the authors would comment on potential effects through CCN1-regulated proteins other than N-cadherin in the discussion.

In fact, other proteins were found regulated in HUVECs upon CCN1 silencing and that could be interesting to investigate further to elucidate other CCN1 functions. The aim of this study though was to identify mechanisms through which CCN1 may be part of the signaling response of cells to increased stiffness. Therefore, we focused on N-Cadherin because it was a protein stiffness regulated (as we have now highlighted in Figure EV1D) and regulated also downstream of CCN1. Conversely, this was not the case for other proteins whose levels were found regulated in siCCN1 cells. Integrin $\alpha 2$, for example, was down-regulated by stiffness. This suggests that CCN1 may regulate Integrin $\alpha 2$ levels but that the pathway involved may be independent of pathways activated by extracellular matrix stiffness. To clarify this point, in the revised manuscript we have integrated the proteomic data obtained from HUVECs cultured at different stiffness with the proteomic data obtained from HUVEC silenced for CCN1 (Figure 3A). This plot clearly highlight that N-Cadherin is one of the few proteins up regulated by stiffness and down regulated when CCN1 is silenced. We decided therefore, not to comment on other proteins regulated upon CCN1 silencing in HUVECs.

Referee #4:

In this manuscript Reid et al study the effect of substrate stiffness on protein expression in endothelial cells, aiming to understand how increased stiffness in the tumour matrix may impact tumour growth and metastasis. To investigate this, the authors use a cell line-based model system (HUVEC cells grown on different substrates) employing SILAC labeling for quantitative proteomics. Functionality of the secreted protein CCN1, one of the proteins that was differentially expressed, was tested in vivo by transplanting B16F10 melanoma cells in CCN1 proficient or deficient mice. This showed that CCN1 contributed to metastasis to the lung in an N-cadherin-dependent manner.

The influence of matrix stiffness (or other properties of the tumor environment) on protein expression cannot be easily assessed in vivo, and therefore the authors resort to a rather artificial in vitro system using a normal endothelial cells line (HUVEC) grown on polyacrylamide gels of different stiffness. The validity of the system is nicely shown in Fig 1a/b. The authors then follow a sound and tested SILAC labeling protocol to examine protein expression, in combination with a clever experiment to correct for effects in protein expression due to differences in proliferation rather than stiffness per se (Fig 1d).

Comments:

1. In the performed SILAC experiment, the authors use rather relaxed cutoff criteria to call differentially expressed proteins (>1 SD from the mean, while usually 2 or 2.5 SD is used) (Fig S1c). In addition, it is not clear what criteria were used to distinguish changes in protein expression that were contributed to stiffness and proliferation, resp (Fig 1E). The figure legends seem to indicate that proteins were required to be regulated in not more than 1 (out of 2) replicate experiments - which is not very strict. Indeed the heat map (left and right panels in Fig 1E) show rather heterogeneous patterns and a considerable number of missing data/ratios especially for the proliferation experiments. Although the authors go on to demonstrate the biological role of CCN1, it is difficult to estimate what fraction of the other proteins mentioned in the figure are deemed to be functional due to the lack of (replicate) data in one or more of the experimental conditions. The authors should more explicitly describe how they treated the data in the methods section rather than in a single sentence in the figure legends.

For the stiffness proteome we have indeed used a quite relaxed threshold, $>1SD$, to be able to identify small yet consistent changes induced by stiffness, thus making our analysis more comprehensive. We used such threshold because we reasoned that SILAC measures differences in protein amounts very accurately and to highlight only consistent changes, we required proteins to be significantly regulated in both replicate experiments.

Regarding the comparison between stiffness and other proliferation datasets, we thank the reviewer for these comments. To address the concerns, we have now performed that analysis in a more solid and unbiased manner. First of all, we have considered for the analysis only proteins that were quantified in all three datasets, stiffness, proliferation on matrigel (soft) and proliferation on plastic (stiff). Then we have Z-scored the data and performed a hierarchical clustering to identify clusters of proteins differently or similarly regulated between datasets. This analysis has identified two distinct

clusters of proteins found more up and down regulated by stiffness than proliferation (Figure 1 E-G). We reported the details of this analysis in the Materials and Methods.

2. Why were different procedures used for proteomic sample preparation (in gel and in solution digestion, page 13)? This might introduce a bias e.g. because proteins elute/digest differently, or because of differences in obtained proteome coverage.

Different procedures were used because different amounts of lysates were collected in the different experiments. For sample preparation, we used the approach that could lead to greater coverage of the proteome (e.g. in-gel for lower amounts of sample and FASP for higher amounts). However, proteins were extracted using the same lysis buffer. For this reason, we believe that, while the number of protein quantified may vary between experiments, for those that have been quantified in all experiments the quantification between samples should be comparable. To avoid any misunderstanding, we have now kept for the comparative analysis only the proteins identified in all the datasets (see above).

3. The authors investigate CCN1 to assess its stiffness-induced expression, however they do not comment on any of the other 'stiffness' proteins (Fig 1E). Is there any pattern/consistency in their known functionality? Or can any of them, with the hindsight of the role and mechanism of CCN1, be placed in the context of the proposed model (Fig 4E)?

We have addressed these questions in two ways:

1. In the main manuscript (Results) we have highlighted that amongst the stiffness regulated proteins there is significant enrichment of proteins involved in cell-cell adhesion and we pinpointed what these proteins are (Figure EV1C,D). These include N-Cadherin/CDH2. This has made clearer the rationale for choosing N-Cadherin for follow up experiments in siCCN1 cells.

2. We have highlighted in the Discussion that amongst the proteins that we have found regulated by stiffness there is a subset that has been previously shown to be under the control of mechanosensitive YAP/TAZ:

“Analysis of the proteome of ECs on hydrogels of physiological and pathological stiffness revealed that ECs respond to high stiffness by upregulating hundreds of proteins. Of those, a subset of 32, which includes CCN1, has been previously shown by chromatin immunoprecipitation assay to be under the control of mechanosensitive YAP/TAZ in MDA-MB-231 breast cancer cells (Zanconato et al, 2015) (Dataset EV1).”

Thank you for submitting your revised manuscript to us. I have now received reports from all original referees, which you can find enclosed below.

As you will see, all referees now support publication, pending satisfactory minor revision. I would thus like to ask you to address the remaining concerns and to provide a further revised manuscript. Referee #1 proposes to remove the *in vivo* data, but both referee #3 and I think that these data should remain. Please make sure, however, that weaknesses are carefully described. Please address all other points raised by referee #1 and #3.

Furthermore, please pay attention to update the following datasets when re-submitting your revised version to better match our publication criteria:

- Please provide a legend in the datasets EV1-3, you can include the legend as a separate tab in the excel document.
- Please zip each movie (EV1 and EV2) together with their respective legend (as separate README doc-files)
- the callout on page 8, line10 (Figure EV2H,J) does not correspond to the EV Figure legends (EV2A-I) and EV Figure (EV2A-I). Maybe the J should be an I? Please clarify.

I am therefore formally returning the manuscript to you for a final round of minor revision. Once we should have received the revised version, we should then be able to swiftly proceed with formal acceptance and production of the manuscript!

 REFEREE REPORTS

Referee #1:

This revised manuscript argues that matrix stiffness affects endothelial cell CCN1 expression and this is linked to the adhesion of cancer cells to endothelial cells. The authors have done a significant amount of work in this newer version; however, the *in vivo* data remains problematic. On reflection, much of the *in vivo* data could be removed and the paper would still be strong.

Specific comments

1. Despite the authors' best efforts the Tat-Cre (HTNC) data remains weak. The authors cannot clearly demonstrate efficient CCN1 deletion by staining for the protein and even the fluorescent reporter strategy is problematic. Further, the appendix figure 4, which apparently includes the tdRFP reporter data, was not included in the manuscript file that I received. The best thing would be to remove these data, especially as the metastasis data are probably not even significant (6/7 vs 3/7 or 4/7 vs 1/7 in Figure 6).
2. There is confusion about the identity of the cancer cells injected in the intravital imaging - the methods suggest B16 cells were used but the supplementary text file states PC3 cells. This needs to be clarified.
3. If there really is a defect in cancer cell interaction with endothelial cells, then what other cells are interacting with the blood vessels or is there more ECM around the blood vessels in the CCN1 endothelial KO?

Referee #2:

The revised manuscript largely addressed the previous concerns when it's experimentally feasible. A few points are not experimentally addressed, but the authors have revised their conclusion statements to be consistent with the data provided. Therefore, the revision is suitable for publication.

Referee #3:

In this manuscript, the authors explore effects of matrix stiffness on the endothelium. They identify Ccn1 as highly upregulated when endothelial cells are cultured on rigid substrates and they demonstrate that increased expression of Ccn1 elevates N-cadherin expression levels in a b-catenin dependent fashion. Interestingly, the Ccn1-mediated upregulation of N-cadherin facilitates cancer cell adhesion to endothelial monolayers in cell culture experiments. By generating endothelial-specific knockout mice, the authors demonstrate that Ccn1 expression levels also impacts N-cadherin expression levels in vivo. Specifically, their data suggest that Ccn1 promotes cancer cell intravasation.

I find this manuscript to be very interesting and the conclusions seem to be supported by the data. As the influence of tissues stiffness on the endothelium is still poorly understood, this manuscripts makes an important contribution that, I think, should be published. Obviously, there are a couple of open questions but answering those would probably go beyond what can be expected from the study at this point.

Minor suggestions:

- the authors should check whether all references to the figures are correct (e.g. I could not find vimentin data in EV3, as indicated in the text).
- Fig.4G does not show any error bars, why is that?
- the authors state that knockdown of b-catenin abolishes Ccn1-mediated upregulation of Cdh2. The data in Fig.4G do not seem to support such a strong interpretation. Cdh2-levels still increase by a factor of two in siCTNNB1 cells. Furthermore, it would be helpful to show overall catenin levels in this blot, so the reader can assess the knockdown efficiency.

Referee #4:

The authors have addressed my concerns, so I recommend the manuscript to be accepted for publication.

2nd Revision - authors' response

08 June 2017

We would like to thank again Editor and Reviewers for their positive comments and for pointing out some minor issues that we have now addressed.

Editor's comments:

- Please provide a legend in the datasets EV1-3, you can include the legend as a separate tab in the excel document.

The legend has now been included in a separated Datasheet within each Excel file.

- Please zip each movie (EV1 and EV2) together with their respective legend (as separate README doc-files)

The README files have now been zipped together with the correspondent movie.

- the callout on page 8, line10 (Figure EV2H,J) does not correspond to the EV Figure legends (EV2A-I) and EV Figure (EV2A-I). Maybe the J should be an I? Please clarify.

Thanks for highlighting this. J should have been indeed an I and it has now been corrected in the manuscript.

Referee #1:

This revised manuscript argues that matrix stiffness affects endothelial cell CCN1 expression and this is linked to the adhesion of cancer cells to endothelial cells. The authors have done a significant amount of work in this newer version; however, the in vivo data remains problematic. On reflection, much of the in vivo data could be removed and the paper would still be strong.

Specific comments

1. Despite the authors' best efforts the Tat-Cre (HTNC) data remains weak. The authors cannot clearly demonstrate efficient CCN1 deletion by staining for the protein and even the fluorescent reporter strategy is problematic. Further, the appendix figure 4, which apparently includes the tdRFP reporter data, was not included in the manuscript file that I received. The best thing would be to remove these data, especially as the metastasis data are probably not even significant (6/7 vs 3/7 or 4/7 vs 1/7 in Figure 6).

following the Editor's suggestion, these data have been kept in the manuscript.

To highlight the weakness of the tdRFP reporter data from Tat-Cre (HTNC)-treated mice, we have modified (in blue) the legend of Appendix Figure S4 as follow (and more details about number of mice were already available in the Materials and Methods):

B Representative immunohistochemistry for RFP (from one of two mice that were treated with HTNC and that showed positive staining using anti-RFP antibody) showing positive staining in tumor blood vessels and lung of HTNC treated *Rosa26^{flSTOP-tdRFP}* B16F10 tumor bearing mice, and absence of staining in BSA treated (CTL) mice. Scale bar = 100 μm . Arrowheads highlight RFP positive cells.

C Pecam1 and RFP immunohistochemistry staining quantification in tumor and lung tissue of one *Rosa26^{flSTOP-tdRFP}* mouse treated with HTNC (B). RFP staining shows that HTNC induced recombination in some regions of the tumor and lung. Pecam1 staining quantification was used to show the total amount of blood vessels in the two tissues. n = number of $\sim 250 \mu\text{m}^2$ regions of the tumor/lung tissue assessed in one analyzed tissue section.

2. There is confusion about the identity of the cancer cells injected in the intravital imaging - the methods suggest B16 cells were used but the supplementary text file states PC3 cells. This needs to be clarified.

We thank the Reviewer for spotting this mistake. The README files have now been corrected.

3. If there really is a defect in cancer cell interaction with endothelial cells, then what other cells are interacting with the blood vessels or is there more ECM around the blood vessels in the CCN1 endothelial KO?

That's indeed an interesting question, but the answer is beyond the scope of this work.

Referee #2:

The revised manuscript largely addressed the previous concerns when it's experimentally feasible. A few points are not experimentally addressed, but the authors have revised their conclusion statements to be consistent with the data provided. Therefore, the revision is suitable for publication.

Referee #3:

In this manuscript, the authors explore effects of matrix stiffness on the endothelium. They identify *Ccn1* as highly upregulated when endothelial cells are cultured on rigid substrates and they demonstrate that increased expression of *Ccn1* elevates N-cadherin expression levels in a β -catenin dependent fashion. Interestingly, the *Ccn1*-mediated upregulation of N-cadherin facilitates cancer cell adhesion to endothelial monolayers in cell culture experiments. By generating endothelial-specific knockout mice, the authors demonstrate that *Ccn1* expression levels also impacts N-cadherin expression levels in vivo. Specifically, their data suggest that *Ccn1* promotes cancer cell intravasation.

I find this manuscript to be very interesting and the conclusions seem to be supported by the data. As the influence of tissues stiffness on the endothelium is still poorly understood, this manuscripts makes an important contribution that, I think, should be published. Obviously, there are a couple of open questions but answering those would probably go beyond what can be expected from the study at this point.

Minor suggestions:

- the authors should check whether all references to the figures are correct (e.g. I could not find vimentin data in EV3, as indicated in the text).

Vimentin data are reported in the MS Dataset EV3 and not Figure EV3.

- Fig.4G does not show any error bars, why is that?

Fig. 4G is a western blot and as such does not contain error bars. Did the reviewer mean a different panel? If Fig. 4B, we did not include error bars because this is the quantification of the representative western blot shown in Fig 4A.

- the authors state that knockdown of b-catenin abolishes Ccn1-mediated upregulation of Cdh2. The data in Fig.4G do not seem to support such a strong interpretation. Cdh2-levels still increase by a factor of two in siCTNNB1 cells. Furthermore, it would be helpful to show overall catenin levels in this blot, so the reader can assess the knockdown efficiency.

We have replaced the word abolished with diminished.

It is indeed true that CDH2 levels still increase by a factor two in siCTNNB1 cells, however it has also to be noted that the levels of β -catenin in siCTNNB1 cells increased when cells were overexpressing CCN1-GFP.

We have now included a panel D in Appendix Figure S2 where we provide a representative western blot for total β -catenin in HUVEC silenced for β -catenin and overexpressing CCN1-GFP. The legend of Appendix Figure S2 and the main manuscript (page 10, line 8) have been updated accordingly.

Referee #4:

The authors have addressed my concerns, so I recommend the manuscript to be accepted for publication.

3rd Editorial Decision

09 June 2017

I am pleased to inform you that your manuscript has been accepted for publication in the EMBO Journal. Congratulations!

Corresponding Author Name: Sara Zanivan

Journal Submitted to: The EMBO Journal

Manuscript Number: EMBOJ-2016-94912R